# A Common Pitfall of Margin-based Language Model Alignment: Gradient Entanglement

Hui Yuan[♠1], Yifan Zeng[♠2], Yue Wu[♠3], Huazheng Wang[4], Mengdi Wang[5], Liu Leqi[♠6]

[1,3,5]Princeton University
[2,4]Oregon State University
[6]The University of Texas at Austin[*†]

## Abstract

Reinforcement Learning from Human Feedback (RLHF) has become the predominant approach for aligning language models (LMs) to be more helpful and less harmful. At its core, RLHF uses a margin-based loss for preference optimization, which specifies the ideal LM behavior only in terms of the difference between preferred and dispreferred responses. In this paper, we identify a common pitfall of margin-based methods—the **under-specification** of ideal LM behavior on preferred and dispreferred responses individually, which results in two unintended consequences as the margin increases: (1) The probability of dispreferred (e.g., unsafe) responses may increase, resulting in potential safety alignment failures. (2) The probability of preferred responses may decrease, even when those responses are ideal. We demystify the reasons behind these problematic behaviors: margin-based losses couple the change in the preferred probability with the gradient of the dispreferred one, and vice versa, often preventing the preferred probability from increasing while the dispreferred one decreases, and thus causing a synchronized increase or decrease in both probabilities. We term this effect, inherent in margin-based objectives, **gradient entanglement**. Formally, we derive conditions for general margin-based alignment objectives under which gradient entanglement becomes concerning: *the inner product between the gradient of preferred log-probability and the gradient of dispreferred log-probability is large relative to the individual gradient norms*. Furthermore, we theoretically investigate why such inner products can be large when aligning language models and empirically validate our findings. Empirical implications of our framework further extend to explaining important differences in the training dynamics of various preference optimization algorithms and suggesting future directions for improvement. [1]

## 1 Introduction

Reinforcement Learning from Human Feedback (RLHF) has become a primary approach for aligning Language Models (LMs) to improve their helpfulness and mitigate harmfulness (Bai et al., 2022; Ouyang et al., 2022; Stiennon et al., 2020). This pipeline typically consists of two stages: supervised fine-tuning (SFT), where demonstration data is used to directly teach the model desirable behaviors, and the reinforcement learning (RL) stage, which uses preference data—comparisons between different responses to the same prompt—to highlight the contrast between chosen and rejected responses, with the goal of helping the model learn distinctions between good and bad behaviors.

In its vanilla form, the RL stage first employs a contrastive loss—based on the margin between the scores of the chosen and rejected responses—to train a reward model, followed by policy optimization methods to fine-tune the LM based on the reward model. Leveraging the structure of the problem, a recent line of work has combined these two steps by directly optimizing the language model using a margin-based preference optimization loss of the following general form (Azar et al.,

---

[*]♠: Leading Contributors. Corresponding to: `huiyuan@princeton.edu`, `leqiliu@utexas.edu`.
[†]Work done in part at Princeton Language & Intelligence.
[1]Code for the paper can be found at https://github.com/HumainLab/Understand_MarginPO.

2024; Ethayarajh et al., 2024; Hong et al., 2024; Meng et al., 2024; Pal et al., 2024; Park et al., 2024; Rafailov et al., 2024; Wu et al., 2024; Xu et al., 2024; Yuan et al., 2024; Zhao et al., 2023):[2]

$$\ell(x, y_w, y_l; \theta) = m(h_w(\log \pi_\theta(y_w|x)) - h_l(\log \pi_\theta(y_l|x))), \quad (1)$$

where for language model $\pi_\theta$, $\log \pi_\theta(y_w|x)$ specifies the log-probability of the chosen response $y_w$ and $\log \pi_\theta(y_w|x)$ specifies that of the rejected response $y_l$[3], given the same prompt $x$. Most of existing preference optimization losses can be interpreted as varying the scalar functions $m, h_w, h_l$ (Section 3.2 and Table 2). At the core, they all rely on the *margin* between the chosen log-probability $\log \pi_\theta(y_w|x)$ and the rejected log-probability $\log \pi_\theta(y_l|x)$.

The training dynamics of these margin-based preference optimization are quite intriguing—the log-probabilities of the chosen and rejected responses often show a synchronized increase and decrease (Figure 1). It is worth noting that, by the end of the training, even though the margin increases (resulting in minimization of the margin-based loss), the log probability of both the chosen and rejected responses may increase (Figure 1a), or both may decrease (Figure 1b).

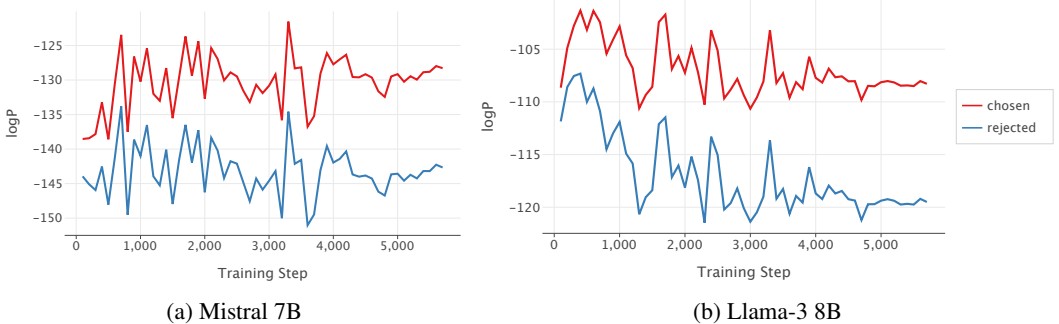

| (a) Mistral 7B | (b) Llama-3 8B |

Figure 1: Training dynamics of the chosen and rejected log probabilities on the TL;DR dataset (Stiennon et al., 2020) during DPO, with log probabilities averaged on the evaluation set. As the margin between the two increases, the chosen and rejected log-probabilities exhibit synchronized increases and decreases per step. In Figure 1a, both chosen and rejected log-probabilities have an overall trend of increasing, especially towards the end of training, whereas in Figure 1b, both have a trend of decreasing. Similar trends are observed across multiple models and datasets, see appendix F).

This synchronized log-probability change exposes a fundamental issue with using margin-based loss for preference optimization in language model alignment: it only specifies the ideal behavior of margin between chosen and rejected log-probabilities, but not the ideal behavior of individual terms. This **under-specification** may have two problematic consequences[4]:

- First, when the primary goal is to reduce the probability of generating rejected responses (e.g., in safety-related alignment tasks where certain undesirable responses should not be generated), merely increasing the margin (i.e., ensuring that the chosen response is preferred over the rejected one) does not guarantee that the log-probability of the rejected response is actually decreasing (Figure 1a).
- Second, even when the log-probability of the rejected response does decrease, the current margin-based losses often lead to a simultaneous reduction in the log-probability of chosen response (Figure 1b). This becomes particularly concerning in some of the current fine-tuning practices where we want to retain or even increase the probability of generating the preferred responses. For example, for distilling strong language models into smaller ones (Chiang et al., 2023; Dubey et al., 2024; Taori et al., 2023; Tunstall et al., 2024), a common practice is to synthesize chosen samples with those strong models; in some alignment applications (e.g., math problem-solving and coding), chosen samples can be the human demonstrations collected during the SFT phase (Chen et al., 2024). In both scenarios, the chosen responses are ideal and we want to ensure the model retains a high probability of generating these ideal responses.

---

[2]The reward modeling loss in vanilla RLHF is also an example of this general form.

[3]Subscript $w$ in chosen response $y_w$ stands for "winner", $l$ in $y_l$ stands for "loser."

[4]In some cases, the ideal LM behavior on chosen and rejected samples can be unclear, e.g., in the original RLHF procedure (Stiennon et al., 2020), the chosen and rejected pairs are drawn from models still in training.

Our study is motivated by the previous two scenarios where, ideally, the LM's probabilities on chosen samples should increase and that on rejected samples should decrease during alignment. However, most margin-based methods fail to induce the ideal behavior (Figure 1, Figure 2), highlighting the need for understanding this common pitfall.

Throughout the paper, we refer to $\log \pi_\theta(y_w|x)$ as the chosen log-probability and its gradient, $\nabla_\theta \log \pi_\theta(y_w|x)$, as the chosen gradient; similar definitions apply for the rejected case. In this work, we *demystify* the reasons why $\log \pi_\theta(y_w|x)$ and $\log \pi_\theta(y_l|x)$ exhibit synchronized increase or decrease during alignment. We uncover that the underlying cause is the **gradient entanglement** effect inherent in margin-based objectives: margin-based losses couple the change in the chosen log-probability with the gradient of the rejected one, and vice versa, preventing the chosen and rejected probabilities from changing independently.

Formally, we characterize gradient entanglement happens because the change in the chosen and rejected probability depends on the inner product $\langle \nabla_\theta \log \pi_\theta(y_w|x), \nabla_\theta \log \pi_\theta(y_l|x) \rangle$ between the chosen and rejected gradients. This entanglement will result in synchronized changes in the chosen and rejected log-probability when the inner product is "large" relative to their individual norms, which we name by "gradient condition" (Section 3.1). Moreover, the precise definitions of "large" for different margin-based algorithms are captured by a general version of the gradient condition (Section 3.2). The gradient conditions we derived enable us to characterize existing margin-based preference optimization methods, explain their differing training dynamics, and identify the most suitable scenarios for deploying these algorithms. Our theoretical findings are also validated through empirical observations (Section 3.3).

We further investigate why the gradient inner product can be relative large to individual norms when aligning a model using language data. In synthetic settings, we theoretically show that (1) as the chosen and rejected responses share more similar tokens, their gradient inner product will increase, and (2) while the sentence-level gradient inner product may be large and positive, individual token-level inner products can be small and negative (Section 4.1.1, 4.1.2). We validate these theoretical insights empirically (Section 4.2). In Section 5, we discuss the empirical implications of our framework for language model alignment.

To summarize, our contributions are as follows:

- We identify a fundamental issue with margin-based alignment: it under-specifies the ideal behavior of the LM on chosen and rejected responses individually (Section 1);
- We uncover that gradient entanglement is the inherent cause of the pitfalls in margin-based objectives, and provide a general gradient inner product condition that captures when the synchronized movement of chosen and rejected log probabilities occurs (Section 3);
- We investigate the gradient inner product and explore when the condition may fail and the synchronized movement occurs theoretically and experimentally (Section 4).

## 2 BACKGROUND AND RELATED WORK

### 2.1 PROBLEM SETUP

We consider auto-regressive language models $\pi(y^t|x, y^{<t})$ that specify the distribution of the next token $y^t$ at index $t$ on a finite vocabulary set $\mathcal{V}$, given the prefix tokens including the prompt $x$ and the partially generated responses $y^{<t}$. In the context of LM alignment, there is a reference policy $\pi_{\text{ref}}$, usually obtained by large-scale pre-training and supervised fine-tuning, and serves as the sampling policy and start point of further alignment algorithms.

### 2.2 PREFERENCE OPTIMIZATION

There have been plenty of works on the design of preference optimization losses, motivated by various assumptions or considerations. Here we briefly review them and discuss their connection to the probability **margin**:

Rafailov et al. (2024) derive the DPO loss from the KL-constrained reward maximization problem:

$$\max_\theta \mathbb{E}_{x \sim \mathcal{X}, y \sim \pi_\theta(\cdot|x)}[r(y; x)] - \beta \mathbb{E}_{x \sim \mathcal{X}}[\text{KL}(\pi_\theta(\cdot|x) \| \pi_{\text{ref}}(\cdot|x))].$$

They further derive the DPO loss for any triplet $(x, y_w, y_l)$ where the $y_w, y_l$ are the chosen and rejected response, respectively:

$$\ell_{\text{DPO}}(x, y_w, y_l; \theta; \pi_{\text{ref}}) := -\log \sigma \left( \beta \left[ \log \left( \frac{\pi_\theta(y_w|x)}{\pi_{\text{ref}}(y_w|x)} \right) - \log \left( \frac{\pi_\theta(y_l|x)}{\pi_{\text{ref}}(y_l|x)} \right) \right] \right). \tag{2}$$

Following the margin-based objective in DPO, advancements have been proposed by IPO (Azar et al., 2024), SlicHF (Zhao et al., 2023), R-DPO (Park et al., 2024), SimPO (Meng et al., 2024), RRHF (Yuan et al., 2024), KTO (Ethayarajh et al., 2024), SPPO (Wu et al., 2024), CPO (Xu et al., 2024) and DPOP (Pal et al., 2024). Pal et al. (2024) is the most relevant work to ours, which touches upon a similar failure mode, but focuses only on the decreasing probability of chosen in DPO. In contrast, we dig deeper to obtain a broader view on the synchronized change (increase or decrease) for a range of margin-based methods, by rigorously analysis and extracting a general success/failure condition. A detailed review of these works is in Appendix B.

## 3 GRADIENT ENTANGLEMENT

Margin-based preference optimization often results in synchronized increase/decrease in chosen and rejected log-probabilities (Section 1). Our key finding is that the synchronized change is caused by an effect we term as gradient entanglement. Starting with a case study on DPO in Section 3.1, we formally define the gradient entanglement effect, from the definition we will see the entanglement is passed through the inner product between chosen and rejected gradients. We derive conditions on such inner product under which the gradient entanglement causes concerning synchronized change. In Section 3.2, we identify gradient entanglement for general margin-based preference optimization methods and apply our framework to explain the training dynamics of those methods. We validate our findings empirically in Section 3.3.

### 3.1 CASE STUDY: GRADIENT ENTANGLEMENT IN DPO

Let us start with deriving the gradient of the DPO objective (2). To simplify the formula of DPO gradient, we define the implicit reward $\hat{r}_\theta(x, y) := \beta \log \frac{\pi_\theta(y|x)}{\pi_{\text{ref}}(y|x)}$ (which is a scalar) and introduce the notations:

$$\log \pi_w(\theta) := \log \pi_\theta(y_w|x), \ \log \pi_l(\theta) := \log \pi_\theta(y_l|x), \ c(\theta) := \sigma\left(\hat{r}_\theta(x, y_l) - \hat{r}_\theta(x, y_w)\right) > 0.$$

Then considering a single sample $(x, y_w, y_l)$, the DPO gradient can be rewritten as[5]

$$\nabla_\theta \ell_{\text{DPO}} = -\beta c(\theta) \cdot (\nabla_\theta \log \pi_w(\theta) - \nabla_\theta \log \pi_l(\theta)). \tag{3}$$

Suppose $\eta > 0$ is the step size for minimizing the DPO objective and let $C = \eta \beta c(\theta)$. After one step gradient descent with (3), a simple analysis of the log-probability change in chosen and rejected responses uncovers the intriguing **gradient entanglement** effect as follows:

> **Gradient Entanglement (DPO)**
>
> The chosen log-probability change $\Delta \log \pi_w$ depends on the rejected gradient $\nabla \log \pi_l$, and similarly, the rejected log-probability change $\Delta \log \pi_l$ depends on the chosen gradient $\nabla \log \pi_w$:
>
> $$\Delta \log \pi_w \approx C \cdot \left( \|\nabla \log \pi_w\|^2 - \langle \nabla \log \pi_w, \nabla \log \pi_l \rangle \right), \tag{4}$$
>
> $$\Delta \log \pi_l \approx C \cdot \left( \langle \nabla \log \pi_w, \nabla \log \pi_l \rangle - \|\nabla \log \pi_l\|^2 \right). \tag{5}$$

(4) and (5) are derived by approximating $\Delta \log \pi_w$ and $\Delta \log \pi_l$ with first-order Taylor expansion (Appendix C.1). Beyond the DPO objective, the gradient entanglement effect is an inherent characteristic of margin-based objectives as the chosen and rejected log-probability are coupled in the definition of "margin." In Section 3.2, we will formally derive gradient entanglement for general margin-based objectives for preference optimization. In the sequel, we will derive conditions on $\langle \nabla \log \pi_w, \nabla \log \pi_l \rangle$ under which the gradient entanglement will have concerning effects.

---

[5]When the context is clear, we omit $\theta$ and just use $\log \pi_w, \log \pi_l$ and $\nabla$.

### 3.1.1 WHEN WILL THE GRADIENT ENTANGLEMENT BE CONCERNING?

If we measure the change in the margin between $\log \pi_w$ and $\log \pi_l$, i.e., the quantitiy $\Delta(\log \pi_w - \log \pi_l)$, then the Cauchy–Schwarz inequality ensures:

$$\Delta(\log \pi_w - \log \pi_l) \approx C \cdot (\|\nabla \log \pi_w\|^2 - 2\langle \nabla \log \pi_w, \nabla \log \pi_l \rangle + \|\nabla \log \pi_l\|^2) \geq 0,$$

which fulfills the contrastive goal of the DPO loss: enlarging the difference between the chosen log-probability $\log \pi_w$ and rejected log-probability $\log \pi_l$. However, due to the gradient entanglement effect, to individually ensure the increment of $\log \pi_w$ and the decrement of $\log \pi_l$, the inner product between chosen and rejected gradient should satisfy the following condition, which we will refer to as "gradient condition".

**Condition 1** (Gradient condition for DPO). *In DPO, to increase $\log \pi_w$ and decrease $\log \pi_l$ individually, (4) and (5) imply the following conditions:*

$$\langle \nabla \log \pi_w, \nabla \log \pi_l \rangle \leq \|\nabla \log \pi_w\|^2 \iff \Delta \log \pi_w \geq 0, \log \pi_w \text{ increases};$$
$$\langle \nabla \log \pi_w, \nabla \log \pi_l \rangle \leq \|\nabla \log \pi_l\|^2 \iff \Delta \log \pi_l \leq 0, \log \pi_l \text{ decreases}.$$

Based on the two conditions above, in Table 1 we summarize three cases that depict all possible changes on the chosen and rejected log-probabilities and are categorized by the value of $\langle \nabla \log \pi_w, \nabla \log \pi_l \rangle$.

| Case | $\Delta \log \pi_w, \Delta \log \pi_l$ | $\log \pi_w, \log \pi_l$ | Condition |
|------|------|------|------|
| 1 | $\Delta \log \pi_w \geq 0 \geq \Delta \log \pi_l$ | $\log \pi_w \uparrow \log \pi_l \downarrow$ | $\langle \nabla \log \pi_w, \nabla \log \pi_l \rangle \leq \min(\|\nabla \log \pi_w\|^2, \|\nabla \log \pi_l\|^2)$ |
| 2 | $0 \geq \Delta \log \pi_w \geq \Delta \log \pi_l$ | $\log \pi_w \downarrow \log \pi_l \downarrow$ | $\|\nabla \log \pi_w\|^2 \leq \langle \nabla \log \pi_w, \nabla \log \pi_l \rangle \leq \|\nabla \log \pi_l\|^2$ |
| 3 | $\Delta \log \pi_w \geq \Delta \log \pi_l \geq 0$ | $\log \pi_w \uparrow \log \pi_l \uparrow$ | $\|\nabla \log \pi_l\|^2 \leq \langle \nabla \log \pi_w, \nabla \log \pi_l \rangle \leq \|\nabla \log \pi_w\|^2$ |

Table 1: Three possible cases of the changes on chosen and rejected log-probabilities in DPO. $\uparrow$ and $\downarrow$ indicate increase and decrease. **Case 1 (Ideal)**: $\log \pi_w$ increases and $\log \pi_l$ decreases; **Case 2**: $\log \pi_w$ and $\log \pi_l$ both decreases but $\log \pi_l$ decreases more; **Case 3**: $\log \pi_w$ and $\log \pi_l$ both increases but $\log \pi_w$ increases more.

### 3.2 GENERAL GRADIENT ENTANGLEMENT EFFECT

We now move on to the general margin-based loss (1). Here, we additionally consider regularizers used in these losses:

$$\ell(\theta) = -\Big( m(h_w(\log \pi_w) - h_l(\log \pi_l)) + \Lambda(\log \pi_w) \Big), \tag{6}$$

where $\Lambda(\log \pi_\theta(y_w|x))$ is a scalar regularizer depending on the chosen log-probability. We instantiate popular preference optimization methods from this general form in Table 2, where we denote $c_{\text{ref}}^w := \log \pi_{\text{ref}}(y_w|x), c_{\text{ref}}^l := \log \pi_{\text{ref}}(y_l|x), c_{\text{ref}} := c_{\text{ref}}^w - c_{\text{ref}}^l$. Terms that only depend on $\pi_{\text{ref}}(y|x)$ shall be viewed as constant, independent of $\theta$.

Based on this unified formulation of preference optimization objectives (6), we derive general gradient entanglement for all margin-based losses (derivations in Appendix C.1):

---

**Gradient Entanglement (General)**

The chosen log-probability change depends on the rejected gradient, and vice versa. The mutual dependency is characterized by:

$$\Delta \log \pi_w \approx \eta \left( d_w \|\nabla \log \pi_w\|^2 - d_l \langle \nabla \log \pi_w, \nabla \log \pi_l \rangle \right),$$
$$\Delta \log \pi_l \approx \eta \left( d_w \langle \nabla \log \pi_w, \nabla \log \pi_l \rangle - d_l \|\nabla \log \pi_l\|^2 \right).$$

---

In the general form of gradient entanglement, $d_w$ and $d_l$ are scalars defined as

$$d_w := m'(h_w(\log \pi_w) - h_l(\log \pi_l))h_w'(\log \pi_w) + \Lambda'(\log \pi_w), \tag{7}$$
$$d_l := m'(h_w(\log \pi_w) - h_l(\log \pi_l))h_l'(\log \pi_l). \tag{8}$$

We derive a generalized version of Condition 1 for general margin-based losses.

**Condition 2** (Gradient condition for general margin-based objectives). *For margin-based preference optimization objectives*(6)*, the conditions for* $\log \pi_w$ *to increase and for* $\log \pi_l$ *to decrease are:*

$$\langle \nabla \log \pi_w, \nabla \log \pi_l \rangle \leq \frac{d_w}{d_l} \|\nabla \log \pi_w\|^2 \iff \Delta \log \pi_w \geq 0, \log \pi_w \ \text{increases}; \qquad (9)$$

$$\langle \nabla \log \pi_w, \nabla \log \pi_l \rangle \leq \frac{d_l}{d_w} \|\nabla \log \pi_l\|^2 \iff \Delta \log \pi_l \leq 0, \log \pi_l \ \text{decreases}. \qquad (10)$$

Accordingly, we can instantiate Condition 2 for different algorithms by using their specialized $m, h_w, h_l, \Lambda$ in Table 2.

| | $m(a)$ | $h_w(a)$ | $h_l(a)$ | $\Lambda(a)$ |
|---|---|---|---|---|
| DPO (Rafailov et al.) | $\log \sigma(a - c_{\text{ref}})$ | $\beta a$ | $\beta a$ | — |
| R-DPO (Park et al.) | $\log \sigma(a - (c_{\text{ref}} + \alpha(|y_w| - |y_l|)))$ | $\beta a$ | $\beta a$ | — |
| SimPO (Meng et al.) | $\log \sigma(a - \gamma)$ | $\frac{\beta}{|y_w|} a$ | $\frac{\beta}{|y_l|} a$ | — |
| IPO (Azar et al.) | $(a - (c_{\text{ref}} + \frac{1}{2\beta}))^2$ | $a$ | $a$ | — |
| RRHF (Yuan et al.) | $\min(0, a)$ | $\frac{1}{|y_w|} a$ | $\frac{1}{|y_l|} a$ | $\lambda a$ |
| SlicHF (Zhao et al.) | $\min(0, a - \delta)$ | $a$ | $a$ | $\lambda a$ |
| CPO (Xu et al.) | $\log \sigma(a)$ | $\beta a$ | $\beta a$ | $\lambda a$ |
| DPOP (Pal et al.) | $\log \sigma(a - c_{\text{ref}})$ | $\beta a - \lambda \max(0, \log c_{\text{ref}}^w - a)$ | $\beta a$ | — |
| KTO (Ethayarajh et al.) | $a$ | $\lambda_w \sigma(\beta a - (\log c_{\text{ref}}^w + z_{\text{ref}}))$ | $\lambda_l \sigma((\log c_{\text{ref}}^l + z_{\text{ref}}) - a)$ | — |
| SPPO (Wu et al.) | $a$ | $(a - \beta^{-1})^2$ | $(a + \beta^{-1})^2$ | — |

Table 2: Instantiation of margin-based preference optimization losses. The constants in these losses satisfy $\beta, \gamma, \delta, \lambda_w, \lambda_l > 0$.

### 3.2.1 HOW DO OTHER MARGIN-BASED METHODS WORK DIFFERENTLY FROM DPO?

Utilizing the gradient condition we derived, we provide in the following a brief discussion on some existing preference optimization algorithms and explain why these algorithms may work differently from DPO under certain settings.

- **DPO**: $\frac{d_w}{d_l} = \frac{d_l}{d_w} = 1$, reproducing the Condition 1 in DPO setting.
- **SPPO**: $\frac{d_w}{d_l} = \frac{\beta^{-1} - \log \pi_w}{\beta^{-1} + \log \pi_l} > 1$[6], where $\beta^{-1}$ is a large constant. Compared with DPO, SPPO loss ensures that it is easier for $\log \pi_w$ to increase based on (9) and harder for $\log \pi_l$ to decrease due to (10).
- **KTO**: $\frac{d_w}{d_l} \propto \frac{\lambda_w}{\lambda_l}$, where $\lambda_w, \lambda_l$ are two hyperparameters in KTO, fine-tuned according to different tasks and datasets. Thus no general conclusion on the chosen/rejected probability change can be made from our conditions.
- **Explicit regularization on chosen log-probability** (**CPO**, **DPOP**[7], **RRHF** and **Slic-HF**): According to the formulas of $d_w$ and $d_l$ in (7) and (8), the negative log-likelihood (NLL) regularizer on chosen responses enlarges $d_w$ while having no influence on $d_l$ as $\Lambda' \geq 0$ and only appears in (7). As a result, larger $\frac{d_w}{d_l}$ makes condition (9) more lenient and thus the chosen log-probability is more likely to increase.
- **Length-normalization** (**SimPO**, **RRHF** and **IPO**): In SimPO, $\frac{d_w}{d_l} = \frac{|y_l|}{|y_w|}$ and condition (9) and (10) can be rewritten as:

$$\left\langle \frac{\nabla \log \pi_w}{|y_w|}, \frac{\nabla \log \pi_l}{|y_l|} \right\rangle \leq \left\| \frac{\nabla \log \pi_w}{|y_w|} \right\|^2; \quad \left\langle \frac{\nabla \log \pi_w}{|y_w|}, \frac{\nabla \log \pi_l}{|y_l|} \right\rangle \leq \left\| \frac{\nabla \log \pi_l}{|y_l|} \right\|^2. \qquad (11)$$

These conditions imply the following: to ensure increasing chosen log-probability while decreasing rejected log-probability, (11) should hold. This is more lenient than the corresponding condition posed for DPO that $\langle \nabla \log \pi_w, \nabla \log \pi_l \rangle \leq \min(\|\nabla \log \pi_w\|^2, \|\nabla \log \pi_l\|^2)$, when the length of chosen and rejected responses is biased so that either the chosen or rejected gradient norm is significantly greater than the other. Therefore, compared to DPO, SimPO leans towards increasing the chosen probability and decreasing that of the rejected when the preference data

---

[6]See Section C.2 for the derivation.

[7]For DPOP, the regularizer is included in its $h_w(a)$ term in Table 2, due to its design to turn on/off the regularizer based on the value of chosen log-probability.

is heavily length-biased. The same reasoning also applies to **RRHF** and **IPO**[8] for their length normalization design.

### 3.3 EMPIRICAL OBSERVATIONS

We conduct experiments on the TL;DR dataset (Stiennon et al., 2020) to showcase the widely-existing phenomenon that the chosen and rejected log-probabilities have synchronized changes during preference optimization. In addition, Figure 1 depicts how different margin-based preference optimization algorithms influence the log-probability of chosen and rejected responses.

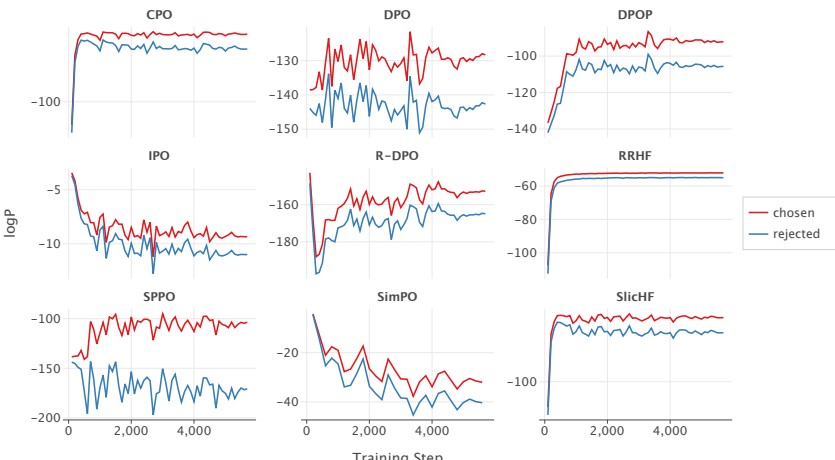

Figure 2: Training dynamics of the chosen and rejected log-probabilities on the TL;DR dataset for different algorithms trained on Mistral 7B. The corresponding plot for Llama3 8B is in Figure 6 (Appendix F). For SimPO and IPO, the log-probabilities are normalized by the response length, while in the other plots, the log-probabilities are of entire responses. All algorithms exhibit synchronized increases and decreases in the chosen and rejected log-probabilities. We also provide the cosine similarity plots between $\nabla_\theta \log \pi_w$ and $\nabla_\theta \log \pi_l$ in Appendix F (Figure 7).

For **DPO** and **R-DPO**, both the chosen and rejected log-probabilities tend to decrease simultaneously. This behavior proves the existence of gradient entanglement, showing that methods purely dependent on the margin might result in both terms decreasing, with the rejected log-probability decreasing more significantly.

**SPPO** demonstrates a distinct trend where the log-probability of the chosen responses increases, while the log-probability of the rejected responses decreases. This matches the theoretical intuition obtained from the specialized gradient conditions for SPPO in Section 3.2.

For **CPO**, **DPOP**, **RRHF**, and **Slic-HF**, algorithms with explicit regularization on the chosen log-probability, we observe a consistent increase in the log-probability of the chosen responses. This behavior reflects the effect of explicit regularizations in increasing the chosen log-probability, which also aligns with the conditions discussed in Section 3.2.

**SimPO** and **IPO**[9] in Figure 1 report the *average* log-probability of responses. Again, an increase in the margin is guaranteed, but not necessarily an increase in the average chosen log-probability due to the gradient entanglement effect.

Overall, experimental results on various margin-based losses closely align with our analysis on the gradient entanglement and the gradient conditions outlined in Section 3.2, demonstrating how loss structures, explicit regularization, length-normalization and other design choices influence the dynamics of preference optimization.

## 4 INVESTIGATION ON GRADIENT INNER PRODUCT

The previous section reveals that the gradient entanglement effect is driven by the key quantity: $\langle \nabla_\theta \log \pi_w, \nabla_\theta \log \pi_l \rangle$ (Condition 1, 2: gradient condition). Margin-based objectives are often trig-

---

[8]In the TRL library, the implementation of IPO averages the log-probabilities by the number of tokens.

[9]In their original paper, Azar et al. (2024) proposed the IPO loss without average log-probability. The authors later claimed using average log-probability with IPO yields improved performance.

gered to not behave in the ideal way, suggesting that the gradient condition is violated due to a large gradient inner product. Therefore, in this section, we investigate into such inner product to understand why it can be large when aligning language models.

Our investigation focuses on the representative margin-based objective DPO and we use toy synthetic settings to analyze this problem and build up our general intuition. All proofs are in Appendix D. Key insights obtained from our analysis are: (1) the gradient inner product increases as the chosen and rejected responses share more similar tokens; and (2) while the sentence-level gradient inner product can be large, individual token-level inner products may be small.[10] We then empirically verify our intuition in Section 4.2.

## 4.1 THEORETICAL RESULTS

### 4.1.1 POSITIVE RESULT ON WHEN THE CONDITION HOLDS

We first provide a positive result when Condition 1 holds and DPO has the ideal behavior that pushes up the log-probability of chosen and pushes down the log-probability of rejected. We begin with set-ups for the LM and preference data.

**Model Setup 1** (**LM with learnable last linear layer**). *Let $V = |\mathcal{V}|$ be the vocabulary size. We assume for prompt $x$ and response $y$, at any index $i$ in the response, the LM outputs:*

$$\pi_\theta(y^i \mid x, y^{<i}) = s(h_i^\top \theta)[y^i],$$

*where $L = |y|$, $\theta \in \mathbb{R}^{d \times V}$ is the learnable parameter, $h_i \in \mathbb{R}^d$ is the hidden state for predicting the $i$-th token in $y$ and $s : \mathbb{R}^V \to \Delta_{\mathcal{V}}$[11] denotes the softmax function. The hidden states are assumed as frozen during DPO.*

**Data Setup 1.** *Both chosen and rejected responses contain only one token under the prompt $x$. That is, $y_w, y_l \in \mathcal{V}^1$, and $y_w[1] \neq y_l[1]$*[12].

The following theorem shows in this task, $\langle \nabla \log \pi_w, \nabla \log \pi_l \rangle < 0$ so that gradient descent steps of DPO make sure $\log \pi_w$ increases and $\log \pi_l$ decreases.

**Theorem 1.** *Under Model Setup 1 and data Setup 1, assume after the SFT stage, given prompt $x$, the model prediction on the first token in response is uniformly concentrated on $M \leq V$ tokens in the vocabulary $\mathcal{V}$, then we have*

$$\langle \nabla \log \pi_w, \nabla \log \pi_l \rangle = -\frac{1}{M}\|h\|^2, \quad \|\nabla \log \pi_w\|^2 = \|\nabla \log \pi_l\|^2 = \frac{M-1}{M}\|h\|^2,$$

*with $h$ being the hidden state for predicting the token that follows prompt $x$. Thus, both parts of Condition 1 hold, resulting in $\log \pi_w$ increases and $\log \pi_l$ decreases.*

Theorem 1 can be extended to the data setup where the chosen and rejected responses have multiple tokens but only differ at the last one, i.e., $y_w[1 : L-1] = y_l[1 : L-1]$, $y_w[L] \neq y_l[L]$ with $L \geq 2$ being the number of tokens in $y_w$ or $y_l$.

**Corollary 2.** *Under Model Setup 1, the chosen and rejected responses only differ at their last token, assume after SFT the model prediction on the $L$-th token in response is uniformly concentrated on $M \leq V$ tokens in the vocabulary, we have $\langle \nabla \log \pi(y_w^L | x, y_w^{<L}), \nabla \log \pi(y_l^L | x, y_l^{<L}) \rangle < 0$, thus at token $L$, the chosen log-probability $\log \pi(y_w^L \mid x, y_w^{<L})$ will increase and rejected counterpart will decrease.*

From the proof of Corollary 2 in Appendix D, though the log-probabilities on the last token behave ideally, it is not guaranteed that the whole chosen response $y_w$ will increase its likelihood and $\log \pi(y_l \mid x)$ will decrease, due to the correlation between $\nabla \log \pi_w$ and $\nabla \log \pi_l$.

### 4.1.2 NEGATIVE RESULT ON WHEN THE CONDITION IS VIOLATED

From the previous results, we can see that the gradient inner product condition is not violated and DPO has the ideal behavior when the chosen and rejected responses differ only at the last token. To gain theoretical insights on what causes the violation of the condition, we level up our previous data setup to the following.

---

[10]To be specific, by token-wise gradient, we mean $\nabla_\theta \log \pi_\theta(y^i | y^{<i})$.

[11]Here, $\Delta$ denote the probability simplex.

[12]For a vector $y$, we use $y[i]$ to denote its $i$-th entry and use $y[i_1 : i_2]$ to denote its entry from $i_1$ to $i_2$.

**Data Setup 2.** *Chosen and rejected responses have an edit distance* $1$ *and the difference appears in the middle of a response, i.e., the chosen and rejected responses* $y_w \in \mathcal{V}^L$ *and* $y_l \in \mathcal{V}^L$ *satisfy* $y_w[1:m-1] = y_l[1:m-1]$, $y_w[m] \neq y_l[m]$, $y_w[m+1:L] = y_l[m+1:L]$ *for* $1 \leq m < L$.

To analyze the optimization steps of DPO under this data setup, we adopt a simpler setting for parameterizing the LM, where the LM has learnable logits.

**Model Setup 2** (**LM with learnable logits**). *We first consider the setting where the LM output follows the structure: For index* $i \in [L]$,

$$\pi_{\overline{\theta}}(\cdot|x, y_w^{<i}) = s_{w,i}, \quad \pi_{\overline{\theta}}(\cdot|x, y_l^{<i}) = s_{l,i},$$

*where* $s_{w,i}, s_{l,i} \in \Delta_{\mathcal{V}}$ *are the probability distributions of the chosen and rejected response at token* $i$, *respectively. We assume the parameterization:* $s_{w,i} = s(\overline{\theta}_{w,i})$ *and* $s_{l,i} = s(\overline{\theta}_{l,i})$ *with* $s$ *being the softmax function, where* $\overline{\theta}_w$ *and* $\overline{\theta}_l$ *are learnable in the model and to which we take the derivatives.*

Because $y_w[1:m-1] = y_l[1:m-1]$, we have that $s_i = s_{w,i} = s_{l,i}$ for $i \in [m]$. Since $s_{w,i}$ and $s_{l,i}$ are predicted by a shared model, they are not independent and one may impose assumptions to characterize the relationship between them. We denote for $i \in [m+1:L]$, $j_i^*$ to be the vocabulary index of token appearing at $y_w[i]$ and $y_l[i]$. As in Pal et al. (2024), we assume that $s_{w,i}[j_i^*] \geq s_{l,i}[j_i^*]$ and $s_{w,i}[j] \leq s_{l,i}[j]$ for $j \neq j_i^*$. Under this assumption, Theorem 3 shows that in this case the log-probability of the chosen and rejected will likely both decrease after one DPO gradient descent step.

**Theorem 3.** *Under Model Setup* 2 *and data Setup* 2, *after one DPO step, the per-token log-probability change in chosen response* $y_w$ *can be characterized with first-order Taylor expansion: for* $i \in [1:m-1]$, *the per-token chosen log-probability before the differing token stays unchanged:*

$$\Delta \log \pi(y_w^i \mid x, y_w^{<i}) \approx 0. \tag{12}$$

*For* $i = m$, *the chosen log-probability at the differing position will increase: suppose* $j^*$ *and* $k^*$ *are the indices of* $y_w[m]$ *and* $y_l[m]$ *in the vocabulary* $\mathcal{V}$,

$$\Delta \log \pi(y_w^m \mid x, y_w^{<m}) \approx 1 + (s_{w,m}[j^*] - s_{w,m}[k^*]) \geq 0. \tag{13}$$

*For* $i \in [m+1:L]$, *the chosen log-probability at these positions will decrease:*

$$\Delta \log \pi(y_w^i \mid x, y_w^{<i}) \approx (1 - s_{w,i}[j_i^*])(s_{l,i}[j_i^*] - s_{w,i}[j_i^*]) - \sum_{j \neq j_i^*} s_{w,i}[j](s_{l,i}[j] - s_{w,i}[j]) \leq 0, \tag{14}$$

*since* $s_{l,i}[j_i^*] - s_{w,i}[j_i^*] \leq 0$ *and* $s_{l,i}[j] - s_{w,i}[j] \geq 0$. *Given the change in sentence-wise log-probability of chosen is the summation of the per-token changes specified in* (12), (13) *and* (14), *as the same suffix following the differing tokens gets longer,* $\log \pi_w$ *decreases more.*

**Remark.** *While Theorem* 3 *adopts the same assumptions made in* Pal et al. (2024), *we precisely characterize the per-token log-probability changes based on the first-order approximation, and explicitly break down the sentence-wise probability change for chosen into* 3 *parts: before/at/after the differing position. Therefore, the analysis in Theorem* 3 *captures the varying probability change directions at different positions, uncovering the underlying dynamic behind the overall decreased chosen probability observed in experiments (Figure* 3).

## 4.2 EMPIRICAL OBSERVATIONS

We verify our intuition regarding when the gradient inner product condition may be held or violated using a sentiment classification task trained on GPT-2, where the prompt $x$ is a statement, e.g., "Happy mothers day mom xoxo." The chosen response $y_w$ specifies the correct sentiment, while the rejected response $y_l$ gives the wrong one. We consider three styles of responses:

- **Single token**: $y_w$: positive. $y_l$: negative.
- **Short suffix**: $y_w$: It has a positive sentiment. $y_l$: It has a negative sentiment.
- **Long suffix**: $y_w$: It has a positive sentiment based on my judgement. $y_l$: It has a negative sentiment based on my judgement.

Empirical observation validates three implications obtained from our theorems:

- First, As showing in Figure 3, the chosen log probability increases only in the **single token** case, aligning with the theoretical prediction by Theorem 1. The **short suffix** chosen log probability decreases less than that of the **long suffix** as responses in **long suffix** contain more tokens following the differing spot, aligning with the theoretical prediction by Theorem 3.
- Taking one step deeper behind the behavior of log-probabilities, the gradient cosine similarity in the **single token** case quickly declines and stays negative during training, while that in the **short suffix** and **long suffix** is positive and increases as the suffix length grows (Figure 4a). This aligns with our gradient condition (Condition 1), where the drop in chosen log probability depends on the magnitude of the gradient inner product.
- Finally, we inspect the token-wise gradient inner product in the **long suffix** case. From the heat map of token-wise gradient similarities (Figure 4b), we observe that on the diagonal, the inner product between the gradients on the tokens "positive" and "negative" is negative, whereas for other identical tokens in the two responses, the gradient cosine similarities are significantly higher and close to 1 for some token pairs.

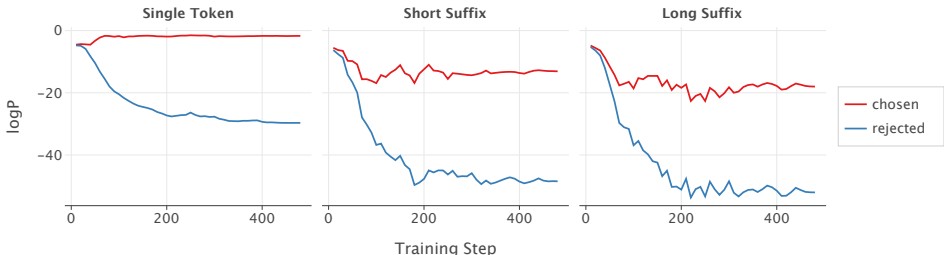

Figure 3: Training dynamics of the chosen and rejected log probabilities for sentiment tasks.

Our theoretical and empirical investigation into the token-level gradient inner product suggests broader implications for general alignment tasks. **Significant tokens** (e.g., "positive"/"negative") contrasting the chosen and rejected responses the most, exhibit negative gradient correlation and prevent gradient entanglement. Meanwhile, those non-contrastive **insignificant tokens** (e.g., identical tokens) cause gradient entanglement due to the high similarity in their gradients. This insight highlights the importance of token-level gradient dynamics and their contribution to the entanglement effect.

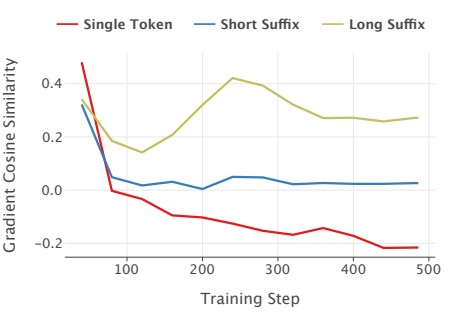
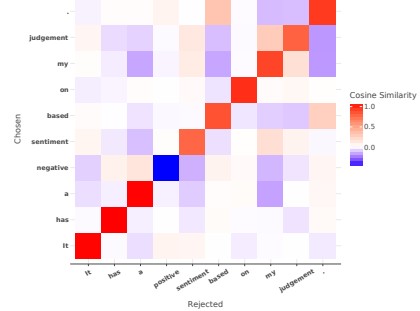

(a) Cosine similarity between $\nabla_\theta \log \pi_w$ and $\nabla_\theta \log \pi_l$.      (b) Token-wise gradient cosine similarity.

Figure 4: Gradient cosine similarity behaviors on the sentence-level and token-level for sentiment tasks. Figure 4a gives the cosine similarity between $\nabla_\theta \log \pi_w$ and $\nabla_\theta \log \pi_l$ for DPO on **single token**, **short suffix** and **long suffix** datasets, defined as: $\frac{\langle \nabla_\theta \log \pi_w, \nabla_\theta \log \pi_l \rangle}{\|\nabla_\theta \log \pi_w\|\|\nabla_\theta \log \pi_l\|}$. Figure 4b shows the token-wise gradient similarity for an instance in the **long suffix** task.

## 5 IMPLICATIONS

In this paper, we touch upon a common pitfall of margin-based preference optimization methods in language alignment. At a high level, our work highlights the need to reconsider the current margin-based preference optimization paradigm. While this approach may enable language models to effectively learn contrasts between good and bad responses, it may not be well-suited for settings where the focus is on the behavior of either the rejected or chosen samples—such as in safety-critical alignment tasks or when distilling from a strong model. See more discussion in Appendix A.

ACKNOWLEDGMENTS

Mengdi Wang acknowledge support by NSF grants DMS-1953686, IIS-2107304, CMMI-1653435, and ONR grant 1006977. Huazheng Wang acknowledges the support by NSF grant IIS-2403401.

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

## A   BROADER IMPLICATIONS

In this paper, we touch upon a common pitfall of margin-based preference optimization methods in language alignment: it underspecifies the ideal behavior of the LM on the chosen and rejected responses individually. Our gradient inner product condition suggests that when the chosen and rejected gradients are similar, their log probabilities will exhibit synchronized increases and decreases. Using this gradient condition, we can categorize existing RLHF variants into two types: (1) those that modify the criterion for the size of the inner product, as seen in the works listed in Table 2, which rely on the same gradient inner product but apply different size criteria; and (2) those that change the inner product of interest directly. As discussed in Section 4, while the sentence-level gradient inner product may be large, the token-level inner product can be small. A line of research, such as advantage-based methods(Mudgal et al., 2023; Setlur et al., 2024), focuses on leveraging token-level information to improve RLHF and falls under the second category.

Finally, at a high level, our work highlights the need to reconsider the current margin-based preference optimization paradigm in language model alignment. While this approach may enable language models to effectively learn contrasts between good and bad responses, it may not be well-suited for settings where the focus is on the behavior of either the rejected or chosen samples—such as in safety-critical alignment tasks or when distilling from a strong model.

## B   REVIEW OF PREFERENCE OPTIMIZATION OBJECTIVES

Motivated by non-transitive human preference and language model calibration respectively, Azar et al. (2024) and Zhao et al. (2023) propose IPO and SlicHF loss with similar forms that solely depend on the **margin** $\log \pi_\theta(y_w|x) - \log \pi_\theta(y_l|x)$.

Due to the length bias observed in practice, Park et al. (2024) propose to add a length penalty term in the BT preference model, but the gradient still relies on the margin $\log \pi_\theta(y_w|x) - \log \pi_\theta(y_l|x)$. Meng et al. (2024) and Yuan et al. (2024) consider the setting of average rewards and derive a loss dependent on the **length-normalized margin** $\frac{1}{|y_w|} \log \pi_\theta(y_w|x) - \frac{1}{|y_l|} \log \pi_\theta(y_l|x)$.

Unlike prior work, Ethayarajh et al. (2024) and Wu et al. (2024) do not consider the difference between the likelihood, but deal with the chosen and rejected response separately. These works typically assign a positive reward signal to the chosen response and a negative reward signal to the rejected one, according to the logistic loss (Ethayarajh et al., 2024) or the square loss (Wu et al., 2024).

(Pal et al., 2024) observes a decrease in the log-probability of chosen response during DPO when the edit distances between each pair of completions are small in preference datasets. To fix the decrease, a natural way is to add explicit regularization to the loss objective, to *force* the increase of the chosen response's log-probability. In particular, (Pal et al., 2024) propose the DPOP loss that behaves the same as DPO when the chosen response's log-ratio $\log \left( \frac{\pi_\theta(y_w|x)}{\pi_{\text{ref}}(y_w|x)} \right)$ is above 0, while adds an explicit regularization when the ratio is below 0. Similarly, Xu et al. (2024) and Zhao et al. (2023) also add explicit regularization to maximize the chosen response's log-probability.

Among these works, the most relevant to ours is Pal et al. (2024), which touches upon a similar failure mode of DPO. The main difference is that they focus on mitigating only the decrease mode of the chosen response's probability by new loss designs. In contrast, we dig deeper to obtain a broader view on the synchronized change (increase or decrease) in chosen and rejected probabilities. We rigorously analyze the training dynamics and extract a general success/failure conditions based on gradient correlation, which applies to a range of margin-based losses for preference optimization.

## C   DERIVATIONS FOR GRADIENT ENTANGLEMENT AND CONDITIONS IN SECTION 3

### C.1   DERIVATION FOR GRADIENT ENTANGLEMENT

**DPO.**   After one step of gradient descent with step size $\eta > 0$ for decreasing the loss $\ell_{\text{DPO}}$, the change in the log-probability of the chosen response denoted by $\Delta \log \pi_w$, as well as the change in the log-probability of the rejected response denoted by $\Delta \log \pi_l$, can be approximated by the

first-order Taylor expansion:

$$\Delta \log \pi_w \approx \langle \nabla_\theta \log \pi_w, -\eta \nabla_\theta \ell_{\text{DPO}} \rangle = \eta \beta c(\theta) \cdot \left( \|\nabla \log \pi_w\|^2 - \langle \nabla \log \pi_w, \nabla \log \pi_l \rangle \right)$$

$$\Delta \log \pi_l \approx \langle \nabla_\theta \log \pi_l, -\eta \nabla_\theta \ell_{\text{DPO}} \rangle = \eta \beta c(\theta) \cdot \left( \langle \nabla \log \pi_w, \nabla \log \pi_l \rangle - \|\nabla \log \pi_l\|^2 \right).$$

**General Losses.** First, the gradient of (6) can be written as

$$\nabla_\theta \ell = d_w \nabla_\theta \log \pi_w - d_l \nabla_\theta \log \pi_l,$$

where $d_w$ and $d_l$ are scalars such that

$$d_w := m'(h_w(\log \pi_w) - h_l(\log \pi_l))h'_w(\log \pi_w) + \Lambda'(\log \pi_w),$$

$$d_l := m'(h_w(\log \pi_w) - h_l(\log \pi_l))h'_l(\log \pi_l).$$

After one step of gradient descend with step size $\eta > 0$ for decreasing the loss $\ell$, the changes in log-probabilities can be approximated by the first-order Taylor expansion:

$$\Delta \log \pi_w \approx \langle \nabla_\theta \log \pi_w, -\eta \nabla_\theta \ell \rangle = \eta \left( d_w \|\nabla_\theta \log \pi_w\|^2 - d_l \langle \nabla_\theta \log \pi_w, \nabla_\theta \log \pi_l \rangle \right),$$

$$\Delta \log \pi_l \approx \langle \nabla_\theta \log \pi_l, -\eta \nabla_\theta \ell \rangle = \eta \left( d_w \langle \nabla_\theta \log \pi_w, \nabla_\theta \log \pi_l \rangle - d_l \|\nabla_\theta \log \pi_l\|^2 \right).$$

### C.2 DERIVATION FOR SPPO

Denote $\mathbf{a} = \nabla_\theta \log \pi(w)$ and $\mathbf{b} = \nabla_\theta \log \pi(l)$. For DPO, we see that the direction of winner and loser is decided by $\langle \mathbf{a}, \mathbf{a} - \mathbf{b} \rangle$ and $\langle \mathbf{b}, \mathbf{a} - \mathbf{b} \rangle$.

Similarly, for any pairwise loss $\ell(\log \pi(w) - \log \pi(l))$, the above statement still holds. Now we take a look at non-pairwise loss $\ell_{\text{SPPO}} = (\log \pi(w) - \beta^{-1})^2 + (\log \pi(l) + \beta^{-1})^2$. We have

$$\frac{d\theta}{dt} = -\nabla_\theta \ell_{\text{SPPO}} = -(\log \pi(w) - \beta^{-1})\nabla_\theta \log \pi(w) - (\log \pi(l) + \beta^{-1})\nabla_\theta \log \pi(l).$$

Then

$$\frac{d}{dt} \log \pi(i) = \left\langle \nabla_\theta \log \pi(i), \frac{d\theta}{dt} \right\rangle$$

$$= -(\log \pi(w) - \beta^{-1})\langle \nabla_\theta \log \pi(i), \nabla_\theta \log \pi(w) \rangle - (\log \pi(l) + \beta^{-1})\langle \nabla_\theta \log \pi(i), \nabla_\theta \log \pi(l) \rangle.$$

We have

$$\frac{d}{dt} \log \pi(w) \approx -(\log \pi(w) - \beta^{-1})\langle \mathbf{a}, \mathbf{a} \rangle - (\log \pi(l) + \beta^{-1})\langle \mathbf{a}, \mathbf{b} \rangle$$

which means if we want $\log \pi(w)$ to increase, we need

$$\frac{\langle \mathbf{a}, \mathbf{b} \rangle}{\langle \mathbf{a}, \mathbf{a} \rangle} < \frac{\beta^{-1} - \log \pi(w)}{\beta^{-1} + \log \pi(l)} =: \alpha.$$

Note that the inequality above implicitly assume that $\beta^{-1} + \log \pi(l) > 0$. This is true in practice as we set $\beta^{-1}$ to be extremely large. Similarly, if we want $\log \pi(l)$ to decrease, we need

$$\frac{\langle \mathbf{a}, \mathbf{b} \rangle}{\langle \mathbf{b}, \mathbf{b} \rangle} < \frac{\beta^{-1} + \log \pi(l)}{\beta^{-1} - \log \pi(w)} =: \alpha^{-1}.$$

We have $\alpha > 1$. It seems SPPO can make sure that $\log \pi(w)$ goes up more easily but also make $\log \pi(l)$ goes up more easily, compared to DPO.

## D PROOFS FOR THE GRADIENT INNER PRODUCT IN SECTION 4

### D.1 LM WITH LEARNABLE LAST LINEAR LAYER: SINGLE TOKEN CASE

We prove Theorem 1 below.

$$\langle \nabla \log \pi_w, \nabla \log \pi_l \rangle = \langle \nabla_\theta \log \pi(y_w^1 \mid x), \nabla_\theta \log \pi(y_l^1 \mid x) \rangle,$$

where $\theta \in \mathbb{R}^{d \times V}$. Let $h \in \mathbb{R}^d$ be the hidden state for the token next to the prompt, $s(\cdot)$ is the softmax function, then

$$\nabla_\theta \log \pi(y_w^1 \mid x) = \nabla_\theta \left( \log s(h^\top \theta)[y_w^1] \right), \tag{15}$$

$$\nabla_\theta \log \pi(y_l^1 \mid x) = \nabla_\theta \left( \log s(h^\top \theta)[y_l^1] \right). \tag{16}$$

Compute the gradient with chain rule,

$$\nabla_\theta \log \pi_w = [-s(1)h, \cdots, (1 - s(i_w))h, \cdots, -s(i_l)h, \cdots, -s(V)h], \tag{17}$$

$$\nabla_\theta \log \pi_l = [-s(1)h, \cdots, -s(i_w)h, \cdots, (1 - s(i_l))h, \cdots, -s(V)h], \tag{18}$$

$i_w, i_l$ are the index of token $y_w^1$ and $y_l^1$ in vocabulary, respectively. For any index $i$, $s(i_w)$ denote LLM's output logit for the $i$-th token in vocabulary.

Suppose at the initialization of $\theta$, $s(1) = \cdots = s(i_w) = \cdots = s(i_l) = s(v) = \frac{1}{M}$ for $M$ entries and the rest $V - M$ entries have they are equal to 0. We note that the exact indices $j$ of which $s(j) = 1/M$ does not matter as it would be the same index for both the chosen and rejected gradients.

$$\nabla \log \pi_w = [-\frac{1}{M}h, \ldots, \underbrace{\left(1 - \frac{1}{M}\right)h}_{i_w - th}, \cdots \underbrace{-\frac{1}{M}h}_{i_l - th}, \cdots, -\frac{1}{M}h], \tag{19}$$

$$\nabla \log \pi_l = [-\frac{1}{M}h, \cdots, \underbrace{-\frac{1}{M}h}_{i_w - th}, \cdots \underbrace{\left(1 - \frac{1}{M}\right)h}_{i_l - th}, \cdots - \frac{1}{M}h], \tag{20}$$

$$\langle \nabla \log \pi_w, \nabla \log \pi_l \rangle = \frac{M-2}{M^2}\|h\|^2 - 2 \cdot \frac{1}{M} \cdot \frac{M-1}{M}\|h\|^2 = -\frac{1}{M}\|h\|^2. \tag{21}$$

$\langle \nabla \log \pi_w, \nabla \log \pi_l \rangle$ is negative. While in comparison, the norms of $\nabla \log \pi_w$ and $\nabla \log \pi_l$ follow:

$$\|\nabla \log \pi_w\|^2 = \|\nabla \log \pi_l\|^2 = \frac{M-1}{M^2}\|h\|^2 + \left(1 - \frac{1}{M}\right)^2 \|h\|^2 = \frac{M-1}{M}\|h\|^2.$$

Therefore, based on Condition 1:

$$\langle \nabla \log \pi_w, \nabla \log \pi_l \rangle = -\frac{1}{M}\|h\|^2,$$

$$\|\nabla \log \pi_w\|^2 = \|\nabla \log \pi_l\|^2 = \frac{M-1}{M}\|h\|^2,$$

$$\log \pi_w \text{ increases and } \log \pi_l \text{ decreases.}$$

## D.2 LM WITH LEARNABLE LAST LINEAR LAYER: MULTI-TOKEN PREFIX CASE

Recall the data setup: the chosen and rejected responses have multiple tokens but only differ at the last one, i.e., $y_w[1 : L-1] = y_l[1 : L-1]$, $y_w[L] \neq y_l[L]$ with $L$ being the length of $y_w$ and $y_l$. We prove Corollary 2 below.

In this case, up to the $L$-th token where chosen and rejected differ, the hidden states are the same for the two responses. This is true because $y_w[1 : L-1] = y_l[1 : L-1]$ and the share the same prompt $x$, so we have that $h_{i,w} = h_{i,l}$ for $i = 1, \cdots, L$, thus we denote both $h_{i,w}$ and $h_{i,l}$ as $h_i$.

For any index $i$, denote $\log \pi_\theta(y_w^i \mid x)$ by $\log \pi_w^i$ and denote $\log \pi_\theta(y_l^i \mid x)$ by $\log \pi_l^i$, then we have

$$\log \pi_w = \sum_{i=1}^{L} \log \pi_w^i, \quad \log \pi_l = \sum_{i=1}^{L} \log \pi_l^i; \tag{22}$$

$$\langle \nabla_\theta \log \pi_w, \nabla_\theta \log \pi_l \rangle = \sum_{i=1}^{L} \sum_{j=1}^{L} \langle \log \pi_w^i, \log \pi_l^j \rangle. \tag{23}$$

Let $h_i \in \mathbb{R}^d$ be the hidden state for predicting the $i$-th token, $s(\cdot)$ is the softmax function, then

$$\nabla_\theta \log \pi_w^i = \nabla_\theta \left( \log s(h_i^\top \theta)[y_w^i] \right),$$

$$\nabla_\theta \log \pi_l^i = \nabla_\theta \left( \log s(h_i^\top \theta)[y_l^i] \right),$$

among which we have $\nabla_\theta \log \pi_w^i = \nabla_\theta \log \pi_l^i$ for $i \in [L-1]$ because $y_w^i = y_l^i$ at those indices. For $i = L$, computing the gradient with chain rule, we have

$$\nabla_\theta \log \pi_w^L = [-s(1)h_L, \cdots, (1 - s(i_w))h_L, \cdots, -s(i_l)h_L, \cdots, -s(V)h_L],$$
$$\nabla_\theta \log \pi_l^L = [-s(1)h_L, \cdots, -s(i_w)h_L, \cdots, (1 - s(i_l))h_L, \cdots, -s(V)h_L].$$

$i_w, i_l$ are the index of token $y_w^L$ and $y_l^L$ in vocabulary, respectively.

Suppose at the initialization of $\theta$, $s(1) = \cdots = s(i_w) = \cdots = s(i_l) = s(v) = \frac{1}{M}$ for $M$ entries and the rest $V - M$ entries have $s(j) = 0$. Similar to the proof of Theorem 1, we have

$$\left\langle \nabla \log \pi_w^L, \nabla \log \pi_l^L \right\rangle = -\frac{1}{M} \|h_L\|^2, \tag{24}$$

$$\|\nabla \log \pi_w^L\|^2 = \|\nabla \log \pi_l^L\|^2 = \frac{M-1}{M} \|h_L\|^2. \tag{25}$$

Therefore, by introducing notations $a_i := \nabla_\theta \log \pi_w^i = \nabla_\theta \log \pi_l^i$ for $i \in [L-1]$, $b_w := \nabla_\theta \log \pi_w^L$ and $b_l := \nabla_\theta \log \pi_l^L$

$$\langle \nabla_\theta \log \pi_w, \nabla_\theta \log \pi_l \rangle = \sum_{i=1}^{L} \sum_{j=1}^{L} \langle \nabla_\theta \log \pi_w^i, \nabla_\theta \log \pi_l^j \rangle$$
$$= \sum_{i=1}^{L-1} \sum_{j=1}^{L-1} \langle a_i, a_j \rangle + \left\langle \sum_{i=1}^{L-1} a_i, b_l \right\rangle + \left\langle \sum_{i=1}^{L-1} a_i, b_w \right\rangle + \langle b_w, b_l \rangle;$$

$$\|\nabla_\theta \log \pi_w\|^2 = \sum_{i=1}^{L} \sum_{j=1}^{L} \langle \nabla_\theta \log \pi_w^i, \nabla_\theta \log \pi_w^j \rangle$$
$$= \sum_{i=1}^{L-1} \sum_{j=1}^{L-1} \langle a_i, a_j \rangle + \left\langle \sum_{i=1}^{L-1} a_i, b_w \right\rangle + \left\langle \sum_{j=1}^{L-1} a_i, b_w \right\rangle + \|b_w\|^2;$$

$$\|\nabla_\theta \log \pi_l\|^2 = \sum_{i=1}^{L} \sum_{j=1}^{L} \langle \log \pi_l^i, \log \pi_l^j \rangle$$
$$= \sum_{i=1}^{L-1} \sum_{j=1}^{L-1} \langle a_i, a_j \rangle + \left\langle \sum_{i=1}^{L-1} a_i, b_l \right\rangle + \left\langle \sum_{i=1}^{L-1} a_i, b_l \right\rangle + \|b_l\|^2;$$

From the equations above, it's ensured that

$$\langle \nabla_\theta \log \pi_w, \nabla_\theta \log \pi_l \rangle < \frac{\|\nabla_\theta \log \pi_w\|^2 + \|\nabla_\theta \log \pi_l\|^2}{2} \tag{26}$$

due to $\langle b_w, b_l \rangle < 0$. However, whether $\langle \nabla_\theta \log \pi_w, \nabla_\theta \log \pi_l \rangle$ will be greater or less than $\min(\|\nabla_\theta \log \pi_w\|^2, \|\nabla_\theta \log \pi_l\|^2)$ depends on the exact absolute value of the term $\langle \sum_{i=1}^{L-1} a_i, \nabla_\theta \log \pi_w^L - \nabla_\theta \log \pi_l^L \rangle$, recall $a_i = \nabla_\theta \log \pi_w^i = \nabla_\theta \log \pi_l^i$. If this absolute value is greater than $\|h_L\|^2$, then $\langle \nabla_\theta \log \pi_w, \nabla_\theta \log \pi_l \rangle > \min(\|\nabla_\theta \log \pi_w\|^2, \|\nabla_\theta \log \pi_l\|^2)$ the condition is violated, otherwise the condition is satisfied. When $L$ is large, in other words, the prefix is long, then the condition is more likely to be violated, leading to the side effect of gradient entanglement.

### D.3 LM with learnable logits setting

We prove Theorem 3 below. We will set up some new notations first. First, we work with the case where $T_w = T_l = L$ is sentence length, $V$ is the vocab size, $y_w[1 : m - 1] = y_l[1 : m - 1]$, $y_w[m] \neq y_l[m]$, and $y_w[m + 1 : L] = y_l[m + 1 : L]$. Note that for all $i \in [L]$, the token $y[i] \in [V]$ is an index, $\bar{\theta}_w$ and $\bar{\theta}_l$ are learnable logits in LM. Each row of the following matrix is $\pi_\theta(\cdot|x, y^{<i}) \in \Delta_{[V]}$ where $i$ is the row index. (Here, there is a slight abuse of notation: $\Delta$ is the probability simplex.) $s : \mathbb{R}^V \to \Delta_V$ is the softmax function.

$$[0,1]^{L \times V} \ni \pi_\theta(x, y_w) = s(\overline{\theta}_w) = \begin{bmatrix} s(\overline{\theta}_w[1, :]) \\ \vdots \\ s(\overline{\theta}_w[m, :]) \\ s(\overline{\theta}_w[m+1, :]) \\ \vdots \\ s(\overline{\theta}_w[L, :]) \end{bmatrix}, \quad \pi_\theta(x, y_l) = s(\overline{\theta}_l) = \begin{bmatrix} s(\overline{\theta}_l[1, :]) \\ \vdots \\ s(\overline{\theta}_l[m, :]) \\ s(\overline{\theta}_l[m+1, :]) \\ \vdots \\ s(\overline{\theta}_l[L, :]) \end{bmatrix} = \begin{bmatrix} s(\overline{\theta}_w[1, :]) \\ \vdots \\ s(\overline{\theta}_w[m, :]) \\ s(\overline{\theta}_l[m+1, :]) \\ \vdots \\ s(\overline{\theta}_l[L, :]) \end{bmatrix}$$

Each row $s(\overline{\theta}[i, :]) \in \Delta_V$. The first $m$ rows are the same for $\overline{\theta}_w$ and $\overline{\theta}_l$ because the tokens up to row $m$ are the same between $y_w$ and $y_l$. The index at row $i$ corresponding to the selected token will be denoted as $j_i^*$, a generic vocab index is $j$. Note that, $j_i^* = j_{i,w}^* = j_{i,l}^*$ for $i \neq m$, and $j_{i,w}^* \neq j_{i,l}^*$ for $i = m$.

Next, the corresponding gradient matrices $\nabla \log s(\overline{\theta}_w), \nabla \log s(\overline{\theta}_l)$ can be specified by:

$$\mathbb{R}^{L \times V} \ni \nabla_\theta \log s(\overline{\theta}_w[i, j_{i+1}^*]) = \begin{bmatrix} \mathbf{0} \\ \vdots \\ \nabla_{\overline{\theta}_w[i,:]} \log s(\overline{\theta}_w[i, j_i^*]) \\ \vdots \\ \mathbf{0} \end{bmatrix}, \quad \nabla_\theta \log s(\overline{\theta}_l) = \begin{bmatrix} \mathbf{0} \\ \vdots \\ \nabla_{\overline{\theta}_l[i,:]} \log s(\overline{\theta}_l[i, j_i^*]) \\ \vdots \\ \mathbf{0} \end{bmatrix}.$$

where

$$\nabla_{\overline{\theta}[i,:]} \log s(\overline{\theta}[i, j_i^*]) \in \mathbb{R}^V, \quad \text{and for } j \in [V], \nabla_{\overline{\theta}[i,:]} \log s(\overline{\theta}[i, j_i^*])[j] = \begin{cases} -s[i, j] & \text{if } j \neq j_i^* \\ 1 - s[i, j] & \text{if } j = j_i^* \end{cases}$$

where $s[i, j] = s(\overline{\theta}[i, :])[j]$, $\log s(\overline{\theta}[i, j_i^*])$ is $j_i^*$-th entry of $\log s(\overline{\theta}[i, :])$, and $\nabla \log s(\overline{\theta}[i, j_i^*])[j]$ is the $j$-th entry of the gradient of $\log s(\overline{\theta}[i, j_i^*])$.

The sentence-wise gradient is

$$\mathbb{R}^{L \times V} \ni \nabla_\theta \mathcal{L} \propto \begin{bmatrix} \nabla \log s(\overline{\theta}_w[1, j_1^*]) - \nabla \log s(\overline{\theta}_w[1, j_1^*]) \\ \vdots \\ \nabla \log s(\overline{\theta}_w[m, j_{m,w}^*]) - \nabla \log s(\overline{\theta}_w[m, j_{m,l}^*]) \\ \nabla \log s(\overline{\theta}_w[m+1, j_{m+1}^*]) - \nabla \log s(\overline{\theta}_l[m+1, j_{m+1}^*]) \\ \vdots \\ \nabla \log s(\overline{\theta}_w[L, j_L^*]) - \nabla \log s(\overline{\theta}_l[L, j_L^*]) \end{bmatrix}$$

$$= \begin{bmatrix} \mathbf{0} \\ \vdots \\ \nabla \log s(\overline{\theta}_w[m, j_{m,w}^*]) - \nabla \log s(\overline{\theta}_w[m, j_{m,l}^*]) \\ \nabla \log s(\overline{\theta}_w[m+1, j_{m+1}^*]) - \nabla \log s(\overline{\theta}_[m+1, j_{m+1}^*]) \\ \vdots \\ \nabla \log s(\overline{\theta}_w[L, j_L^*]) - \nabla \log s(\overline{\theta}_l[L, j_L^*]) \end{bmatrix}$$

Now, let's first derive the token-wise condition for the selected token (learning rate $\eta = 1$):

**Chosen response: if $i = m$, we have**

$$\Delta \log s(\overline{\theta}_w[i, j^*_{i,w}]) \approx \sum_{i'=1}^{L} \langle \nabla \log s(\overline{\theta}_w[m, j^*_{m,w}]), \nabla \mathcal{L}[i', :] \rangle = \langle \nabla \log s(\overline{\theta}_w[m, j^*_{m,w}]), \nabla \mathcal{L}[m, :] \rangle$$

$$= \langle \nabla \log s(\overline{\theta}_w[m, j^*_{m,w}]), \nabla \log s(\overline{\theta}_w[m, j^*_{m,w}]) - \nabla \log s(\overline{\theta}_w[m, j^*_{m,l}]) \rangle$$

$$= \left( \sum_{j' \neq j^*_{m,w}} s_w[m, j']^2 \right) + (1 - s_w[m, j^*_{m,w}])^2$$

$$- \left( \sum_{j' \neq j^*_{m,w}, j' \neq j^*_{m,l}} s_w[m, j']^2 \right) + s_w[m, j^*_{m,w}](1 - s_w[m, j^*_{m,w}]) + s_w[m, j^*_{m,l}](1 - s_w[m, j^*_{m,l}])$$

$$= 1 + (s_w[m, j^*_{m,l}] - s_w[m, j^*_{m,w}]) \geq 0, \tag{27}$$

where the last inequality is true because $s \in [0, 1]$. Here, basically, this margin loss will just encourage increase the chosen logP (and reduce the rejected one) for the selected token.

**Chosen response: if $i \neq m$, we have**

$$\Delta \log s(\overline{\theta}_w[i, j^*_{i,w}]) \approx \sum_{i'=1}^{L} \langle \nabla \log s(\overline{\theta}_w[i, j^*_i]), \nabla \mathcal{L}[i', :] \rangle = \langle \nabla \log s(\overline{\theta}_w[i, j^*_i]), \nabla \mathcal{L}[i, :] \rangle$$

$$= \langle \nabla \log s(\overline{\theta}_w[i, j^*_i]), \nabla \log s(\overline{\theta}_w[i, j^*_i]) - \nabla \log s(\overline{\theta}_l[i, j^*_i]) \rangle$$

$$= (1 - s_w[i, j^*_i])(s_l[i, j^*_i] - s_w[i, j^*_i]) - \sum_{j' \neq j^*_i} s_w[i, j'](s_l[i, j'] - s_w[i, j']) \tag{28}$$

Here, basically, the loss can only pick one direction to change both chosen and rejected entry.

**Connection to the derivation in Pal et al. (2024).** The assumption in Pal et al. (2024) mainly ensures the sign of (28). Basically, smaug's assumption ensures that for $i \in [m + 1, L]$, $s_w[i, j^*_i] \geq s_l[i, j^*_i]$ and $s_w[i, j] \leq s_l[i, j]$ for $j \neq j^*_i$.

$$\nabla \log s(\overline{\theta}_w[i, j^*_i]) - \nabla \log s(\overline{\theta}_l[i, j^*_i]) = \begin{bmatrix} s_l[i, 1] - s_w[i, 1] \\ \vdots \\ s_l[i, j^*_i] - s_w[i, j^*_i] \\ \vdots \\ s_l[i', V] - s_w[i', V] \end{bmatrix} = \begin{bmatrix} \geq 0 \\ \vdots \\ \leq 0 \\ \vdots \\ \geq 0 \end{bmatrix}$$

For (28), we have

$$(1 - s_w[i, j^*_i])(s_l[i, j^*_i] - s_w[i, j^*_i]) - \sum_{j' \neq j^*_i} s_w[i, j'](s_l[i, j'] - s_w[i, j']) \leq 0.$$

This ensures the chosen token will have reduced logP.

**Condition on chosen tokens increasing and rejected token decreasing at $m$, and on chosen and rejected tokens decreasing after $m + 1$:**

$$(27) \geq 0 \text{ always holds,}$$
$$\forall i \in [m + 1, L], \ s_w[i, j^*_i] \geq s_l[i, j^*_i], \ \forall j \neq j^*_i, s_w[i, j] \leq s_l[i, j] \implies (28) \leq 0$$

## E    EXPERIMENT DETAILS

### E.1    HARDWARE AND SOFTWARE SETUP

Our experiments were implemented using TRL version 0.11.0. The training was performed on a hardware setup consisting of two NVIDIA H100 GPUs, providing substantial computational power for the training process.

### E.2    TL;DR TASK SETUP

For the TL;DR summarization task, we utilized the CarperAI/openai_summarize_comparisons dataset. We employed two LLMs for this task:

- mistralai/Mistral-7B-Instruct-v0.3 (referred to as Mistral 7B)
- meta-llama/Meta-Llama-3-8B-Instruct (referred to as Llama-3 8B)

We did not perform any supervised fine-tuning step prior to the RLHF training for these models.

To optimize the training process, we applied Low-Rank Adaptation (LoRA) with a rank of 64 to both models. The learning rate was set at $5 \times 10^{-6}$ for all RLHF training.

### E.3    RLHF ALGORITHM CONFIGURATIONS

We implemented several RLHF algorithms, each with its own specific configurations:

- Direct Preference Optimization (DPO): $\beta = 0.1$
- Chosen NLL term (used in CPO, RRHF, and SLiC-HF): $\lambda = 1$
- SLiC-HF: $\delta = 1$
- SimPO: $\gamma = 0.5$
- R-DPO: $\alpha = 0.2$
- DPOP: $\lambda = 50$

### E.4    SENTIMENT ANALYSIS TASK SETUP

For the sentiment analysis task, we used a specially curated sentiment dataset. Unlike the TL;DR task, we performed supervised fine-tuning on the GPT-2 model before proceeding with the RLHF training. The learning rate for this RLHF training was also set to $5 \times 10^{-6}$.

## F    ADDITIONAL EMPIRICAL RESULTS

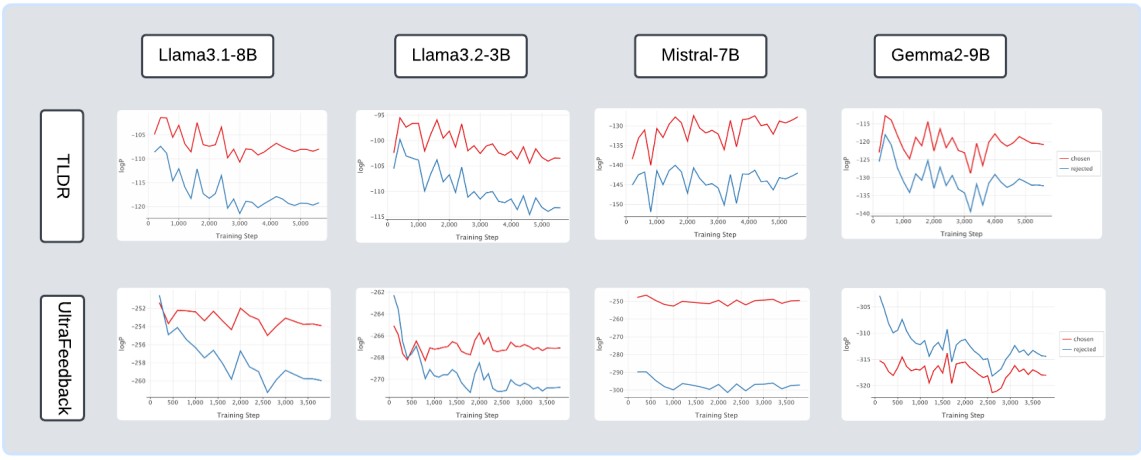

Figure 5: Training dynamics of the chosen and rejected log probabilities during DPO, observed across models: Llama3.1-8B (Dubey et al., 2024), Llama3.2-3B, Mistral-7B (Jiang et al., 2023) and Gemma2-9B (Team et al., 2024) on TL;DR (Stiennon et al., 2020) and UltraFeedback (Cui et al., 2024) datasets. Log probabilities are averaged on the evaluation splits.

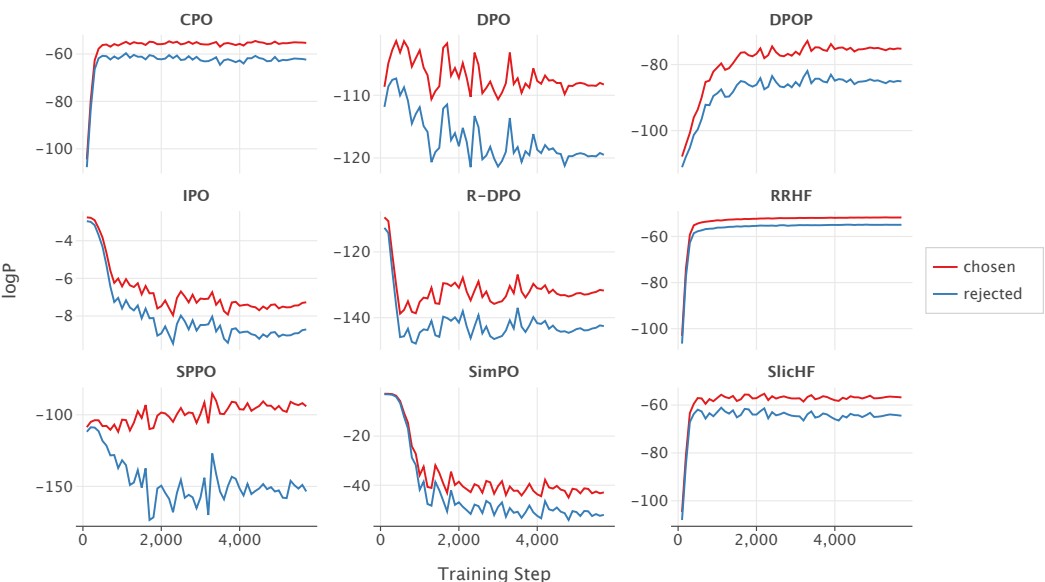

Figure 6: Training dynamics of the chosen and rejected log probabilities on the TL;DR dataset for different preference optimization algorithms trained on Llama-3 8B. All algorithms exhibit synchronized increases and decreases in the chosen and rejected log probabilities. Note: For SimPO and IPO, the log probabilities are normalized, while in the other plots, they are the original log probabilities.

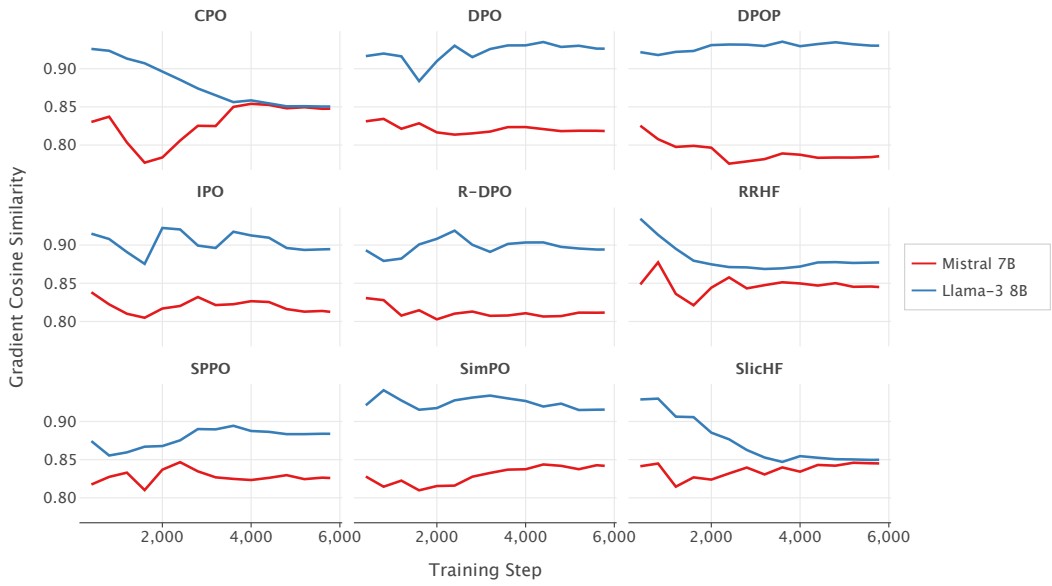

Figure 7: Cosine similarity between $\nabla_\theta \log \pi_w$ and $\nabla_\theta \log \pi_l$ on the TL;DR dataset for different preference optimization algorithms trained on Llama-3 8B and Mistral 7B.

