# OpenReview forum: "A Common Pitfall of Margin-based Language Model Alignment: Gradient Entanglement"
_ICLR.cc/2025/Conference — ICLR 2025 Poster_

### Official Review · Reviewer_ai3q · 2024-10-28

**Soundness:** 3
**Presentation:** 3
**Contribution:** 2
**Rating:** 5
**Confidence:** 3

**Summary:**

This paper criticizes margin-based preference optimization for LM alignment, which is a core method in RLHF. While increasing preference margins can reduce unsafe responses, it also often reduces preferred response probabilities, due to synchronized probability shifts between chosen and rejected responses. The authors reveal when these shifts occur and suggest that the under-specification of behaviour in current margin-based methods limits model safety and effectiveness. The work proposes fine-grained, token-level approaches as potential improvements, though this is not really pursued very deeply in this work.

**Strengths:**

- The paper provides a fairly nuanced and theoretically grounded critique of margin-based preference optimization, which is an increasingly relevant task (especially  in RL). Identifying conditions under which preference optimization can backfire is important for real-world use and may provide a relatively underdeveloped area of research
- The proposal to address preference optimization problems at the token level is interesting (though perhaps not surprising).

**Weaknesses:**

- this is not meant to be a primarily empirical paper (apparently, it leans more on its theoretical aspects) but this work of course calls for broader empirical validation than it has. The proposed method may be evaluated across diverse tasks such as summarization, dialogue, and instruction-following. Other variants of the given LLMs (e.g., LLaMa) could have been explored, to demonstrate generalizability.
- The analysis of Sec 4 focuses on synthetic, controlled scenarios. The kinds of complexities exist that in actual, diverse language data should be addressed.

**Questions:**

- While the theoretical analysis of gradient inner products and synchronized probability shifts is fairly rigorous, what actionable insights for real-world RLHF applications are achievable?
- How would you evaluate the ‘safety-critical alignment tasks’ you mention in Sec 5 as being particularly ill-suited for margin-based preference optimization?
- Can you clarify more directly how token-level gradient inner products between chosen and rejected responses provide a reliable indicator for alignment behaviour?

---

> ### Author Response · Authors · 2024-11-23
> **Response to Reviewer ai3q**
>
> Thank you for acknowledging the importance of our work:
> > "Identifying conditions under which preference optimization can backfire is important for real-world use and may provide a relatively underdeveloped area of research."
>
> and appreciating our work for its being theoretically grounded and interesting. We address your concerns individually as follows.
>
> >**Weakness 1:** This work calls for broader empirical validation than it has. (1) Other variants of the given LLMs (e.g., LLaMa) could have been explored. (2) The proposed method may be evaluated across diverse tasks such as summarization, dialogue, and instruction-following.
>
> **Response to Weakness 1:** Thanks for your suggestions on experiments.
> - To resolve your concern(1), we extended our experiments to cover datasets: TL;DR, UltraFeedback and models:  Llama3.1-8b, Llama3.2-3b, Mistral-7b, Gamma2-9b. Among all combinations of datasets and models, empirical observations are consistently aligned with our theoretical framework, here is the [link](https://anonymous.4open.science/r/Rebuttal_ICLR-12D0/README.md) to plots of new results and more interpretations can be found in the "**New Experiments**"[(link)](https://openreview.net/forum?id=YaBiGjuDiC&noteId=0BOg83tgvw) section in our global rebuttal.
> - For concern(2), we want to kindly remind you that our paper did not propose new methods thus requiring method evaluation may not be valid. The toy synthetic settings including the analysis and experiments we provided in Sec 4 is for building up our general intuition on why the gradient inner product can be large and thus causes the problematic behaviors in chosen and rejected logps during training, rather than for motivating new methods.
>
>
> >**Weakness 2:** The analysis of Sec 4 focuses on synthetic, controlled scenarios. The kinds of complexities exist that in actual, diverse language data should be addressed.
>
> **Response to Weakness 2:** We emphasize that the purpose of Sec 4 is for building up a high-level intuition on why the gradient inner product can be large during alignment. The alignment process on real-world preference data is too complicated to have theoretical analysis, where the variety of language data is a main source of complications. Though being simple, the current data configuration captures fundamental problems in many preference datasets: between the chosen and rejected responses, many times, most of the tokens share similar meanings and only a few tokens carry distinguishing information to separate out chosen and rejected responses. Theoretically extend to some less constrained settings can be an interesting future direction. One may consider the setting where blocks of their tokens are paraphrase to each other that share similar meanings instead of the exact same tokens. In "**Clarification on Theoretical Simplifications**"[(link)](https://openreview.net/forum?id=YaBiGjuDiC&noteId=2z1YkEjPtC)  section in our common rebuttal, you can find more explanations of our theory.

---

> > ### Author Response · Authors · 2024-11-26
> > **Response to Reviewer ai3q #2**
> >
> > **Response to Other Questions**
> >
> > We apologize that the second half of our reply for your other questions are missing previously. Please refer to what follows.
> >
> > >**Q1:** While the theoretical analysis of gradient inner products and synchronized probability shifts is fairly rigorous, what actionable insights for real-world RLHF applications are achievable?
> >
> > **A1:**  There are multiple actionable insights one can derive from our analysis. First, people should be highly aware of the under-specification issue of the current preference optimization objectives in LLM's probability on chosen/rejected samples. When applying preference optimization methods like DPO(*-PO), etc, instead of only focusing on the "reward accuracy" metric that evaluates how well the margin between chosen/rejected samples are enlarged, people should do more well-rounded evaluations on the model behavior during alignment practice.
> >
> > Second, on the algorithmic side, there are also two new potentially beneficial designs of RLHF that can be derived from our analysis:
> > - (1) Reweighing Margin-based Objectives:
> >     We can reweigh the chosen and rejected log-probabilities in the margin-based loss such that ${d_w}/{d_l} = \|\nabla \log \pi_l\| / \|\nabla \log \pi_w\|$, which ensures that both parts in the genral gradient conditons (Condtion2 in Sec 3.2) are satisfied at the same time.
> >
> > - (2) Leveraging Token-level Distinctions:
> >     We can consider picking out certain tokens in both chosen and rejected samples and only apply margin-based loss on those token. In this way, if the token-level gradient product on those picked-out tokens are small or negative, then the synchronized update will not happen. Here we give an example loss following this idea:
> >
> >     $-\log\sigma  \left( \sum_{i=1}^L \mathbb{I}\set{u_w^i \geq r }\log \frac{\pi_{\theta}(y^i_w|x, y_w^{<i})}{\pi_\text{ref}(y^i_w|x, y_w^{<i})}- \mathbb{I}\set{u_l^i \geq r } \log \frac{\pi_{\theta}(y^i_l|x, y_w^{<i})}{\pi_\text{ref}(y^i_l|x, y_l^{<i})} \right) + \eta \left(\|\mathbb{I}\set{u_w \geq r}\|_1 + \|\mathbb{I}\set{u_l \geq r}\|_1\right),$
> >
> >     where $\eta\in \mathbb{R}_+, r \in \mathbb{R}$ are hyper-parameters and $u_w \in \mathbb{R}^L, u_l\in \mathbb{R}^L$ are learnable weights depending on $(x,y_w^{<i}), (x,y_l^{<i})$ respectively, interpreted as the confidence in considering token $i$ significant. The loss is inspired by sparsity-related ideas (e.g., LASSO), The $\ell_1$ regularizer on the token-wise mask imposes sparsity on it.
> >
> > >**Q2:** How would you evaluate the ‘safety-critical alignment tasks’ you mention in Sec 5 as being particularly ill-suited for margin-based preference optimization?
> >
> > **A2:** As we know form our paper, due to the synchronized probability shift, the log-proability of chosen samples can possibly decrease and the log-proability of rejected sample can increase (Figure 1a). So when the rejected samples are harmful and strictly non-allowable for the ideal model behaviors, such as in safety alignment tasks, for the model to have its likelihood/probability on the rejected sample increased can be very concerning.
> >
> > >**Q3:** Can you clarify more directly how token-level gradient inner products between chosen and rejected responses provide a reliable indicator for alignment behavior?
> >
> > **A3:** As shown in the heatmap in Fig4(b), the most contrastive token (e.g., “positive”/“negative”) exhibits negative gradient correlation. Meanwhile, those non-contrastive tokens (e.g., identical tokens) cause gradient entanglement due to the high similarity in their gradients. Based on our (sentence-level) gradient condition, together with the logp of the rejected response, the logp of the entire chosen response will have decreasing logp due to the positive gradient inner product accumulated on those non-contrastive tokens. This is validated empirically in Fig3 (Long suffix). We want to clarify that the token-level gradient inner product itself (such as the inner product (i,j) calculated between token i in chosen and token j in rejected) cannot serve as an indicator of alignment behavior for common margin-based alignment methods like DPO, because in these algorithms the logp change will depend on the inner product of all token pairs. However, the distinctions in token-level gradients shown by the heatmap in Fig4(b) suggests that future algorithms can potentially leverage the token-level distinction to combat gradient entanglement, and for those new algorithms leveraging token-level information differently from that of DPO, the token-level gradient inner products can serve as an indicator of alignment behavior.

---

> ### Author Response · Authors · 2024-11-29
> **Follow-up on Our Rebuttal**
>
> Dear Reviewer ai3q,
>
> We want to remind you that our rebuttals have been posted for a while, it's now approaching the end of the disscussion period. we are eager to hear your thoughts and any further feedback you might have.

---

> ### Author Response · Authors · 2024-12-02
> **Last-Day to Reply: Follow-Up on Our Rebuttal**
>
> Dear Reviewer ai3q,
>
> As the discussion period is coming to a close, we would greatly appreciate it if you could let us know whether our rebuttal has fully addressed your concerns and questions.
> Regarding the two concerns and three questions listed in the review, we provided point-by-point responses, including:
> - **Added Extensive Experiments**: the key phenomenon (synchronized shifts of chosen and rejected log-probabilities) investigated in this paper is widely observed for models: Llama3.1-8b, Mistral-7b, Gamma2-9b, Llama3.2-3b; and commonly used RLHF datasets: TL;DR, UltraFeedback. And the cause we identified (correlated gradient inner products) is verified in all settings.
> - **Clarifying and Justifying Theoretical Simplifications**: We have clarified that the contributions of our theory are two-folded. The first half reveals the cause of the phenomenon, in which no assumptions are made. The second half is intended for gaining intuition, thus some simplifications are made. For this part, we explained how our theoretical settings capture the main characteristics of RLHF data, and discussed how our theoretical findings are generalized empirically in our paper.
> - **Actionable Items**: We have proposed two actionable items (i.e., new learning objectives) in solving the simultaneous shifts problems using our theoretical framework.
> - **New Experiment to Show the Risk of the Identified Pitfall**: We have added new experimental results to showcase how the synchronized shifts can potentially result in unreliable language model behavior in safety-critical alignment tasks.
> - **Token-Level Information in Alignment**: We have provided additional discussion on how token-level information can be considered for language model alignment, and how our empirical results are pointing to this.
>
> Thank you for your time and consideration.
>
> Best regards,\
> Authors of Submission 13302

---

### Official Review · Reviewer_UXg4 · 2024-10-31

**Soundness:** 3
**Presentation:** 2
**Contribution:** 2
**Rating:** 5
**Confidence:** 4

**Summary:**

The paper identifies and analyzes a fundamental problem with margin-based preference optimization in language model alignment (like RLHF): these methods only specify how much better preferred responses should be compared to dispreferred ones, without specifying how the individual probabilities should behave. This leads to two unintended consequences: (1) sometimes the probability of dispreferred (e.g., unsafe) responses increases, and (2) when reducing dispreferred response probabilities, the probability of preferred responses often decreases too, even when they are ideal.

**Strengths:**

The paper addresses a fundamental issue in language model alignment through RLHF by analyzing the under-specification problem in margin-based preference optimization - this is an important contribution given how widely RLHF is used.

**Weaknesses:**

1  Presentation Issues:
The notation becomes dense and potentially confusing in Section 3.2, particularly around the general form analysis
Some figures (like Figure 4b) would benefit from more detailed explanations of their implications
The connection between the theoretical results and practical implications could be more clearly articulated

2  Limited Experimental Validation:
Experiments are conducted on only one main dataset (TL;DR) and a simple sentiment classification task
Only tested on two main models (Mistral 7B and Llama-3 8B)
Lacks validation across diverse preference optimization scenarios and task types
More extensive empirical validation across different datasets and model architectures would strengthen the claims

3  Theoretical Assumptions:
The theoretical analysis in Section 4.1 relies on somewhat simplified model settings (learnable last linear layer, learnable logits)
Some assumptions about token probability relationships between chosen/rejected responses may not hold in practice
The extension of single-token results to multi-token cases could be more rigorously justified

**Questions:**

N/A

---

> ### Author Response · Authors · 2024-11-23
> **Response to Reviewer UXg4**
>
> We thank you for appreciating the importance of our work:
> >"This paper addresses a fundamental under-specification problem in margin-based preference optimization -- this is an important contribution given how widely RLHF is used."
>
> We address your concerns individually as follows.
>
> > **Presentation:** (1) The notation becomes dense and potentially confusing in Section 3.2, particularly around the general form analysis. (2) Some figures (like Figure 4b) would benefit from more detailed explanations of their implications. (3) The connection between the theoretical results and practical implications could be more clearly articulated.
>
> **Our Response:** Thanks for your suggestions on presentation. We have made revisions accordingly, where we (1) simplified the math presentation in Sec 3.2 for general objectives; (2) revised the interpretation of experiment plots including Fig 2, Fig 3 and Fig 4; and (3) strengthened our explanation on the connection between experiments and theory in Sec 3.3 and Sec 4.2. For a More detailed summary of our paper revision, please refer to the "**Paper Revisions**"[(link)](https://openreview.net/forum?id=YaBiGjuDiC&noteId=PYCtGWuToZ) section in our common rebuttal and feel free to check out our revised paper.
>
> > **Limited Experimental Validation:** Experiments are conducted only on TL;DR dataset in addition to a simple sentiment classification task, and only on two main models (Mistral 7B and Llama-3 8B).
>
> **Our Response:** Thanks for your suggestions on experiments. We extended our experiments to cover datasets: TL;DR, UltraFeedback and models:  Llama3.1-8b, Llama3.2-3b, Mistral-7b, Gamma2-9b. Among all combinations of datasets and models, empirical observations are consistently aligned with our theoretical framework, here is the [link](https://anonymous.4open.science/r/Rebuttal_ICLR-12D0/README.md) to plots of new results and more interpretations can be found in the "**New Experiments**"[(link)](https://openreview.net/forum?id=YaBiGjuDiC&noteId=0BOg83tgvw) section in our common rebuttal.
>
> > **Theoretical Assumptions:** The theoretical analysis in Section 4.1 relies on somewhat simplified model settings (learnable last linear layer, learnable logits). Some assumptions about token probability relationships between chosen/rejected responses may not hold in practice. The extension of single-token results to multi-token cases could be more rigorously justified.
>
> **Our Response:**
> - We want to clarify that the synthetic settings in Section 4.1 is for building up our general intuition on why the gradient inner product can be large and thus causes the problematic behaviors in logps. We acknowledge that this setting is toy but the insight obtained from our analysis is non-trivial. In the "**Clarification on Theoretical Simplifications**" [(link)](https://openreview.net/forum?id=YaBiGjuDiC&noteId=2z1YkEjPtC) section in our common rebuttal, you can find more explanations on why we made some theoretical simplifications and a detailed list of the practical implications of our theory. Furthermore, in Section 4, our theoretical findings are verified when fine-tuning full parameters of GPT-2, extending our findings beyond simplified model settings (learnable last layer/logits).
> - For extension of single-token results to multi-token case: we have revised Corollary 2 and provided a rigorous proof for it in Appendix, feel free to check it in our revised version.

---

> > ### Author Response · Authors · 2024-11-29
> > **Follow-up on Our Rebuttal**
> >
> > Dear Reviewer UXg4,
> >
> > We want to remind you that our rebuttals have been posted for a while, it's now approaching the end of the disscussion period. we are eager to hear your thoughts and any further feedback you might have.

---

> ### Author Response · Authors · 2024-12-02
> **Last-Day to Reply: Follow-Up on Our Rebuttal**
>
> Dear Reviewer UXg4,
>
> As the discussion period is coming to a close, we would greatly appreciate it if you could let us know whether our rebuttal has fully addressed your concerns and questions.
> Regarding the two concerns and three questions listed in the review, we provided point-by-point responses, including:
>
> - **Paper Revision**: We have revised the paper to provide easy-to-follw theoretical results, and a clear discussion on the connection between theoretical results and empirical investigation.
> - **Added Extensive Experiments**: the key phenomenon (synchronized shifts of chosen and rejected log-probabilities) investigated in this paper is widely observed for models: Llama3.1-8b, Mistral-7b, Gamma2-9b, Llama3.2-3b; and common RLHF datasets: TL;DR, UltraFeedback.  And the cause we identified (correlated gradient inner products) is verified in all settings.
> - **Clarifying and Justifying Theoretical Simplifications**: We have clarified that the contributions of our theory are two-folded. The first half reveals the cause of the phenomenon, in which no assumptions are made. The second half is intended for gaining intuition, thus some simplifications are made. For this part, we explained how our theoretical settings capture the main characteristics of RLHF data, and discussed how our theoretical findings are generalized empirically in our paper.
>
>
> Thank you for your time and consideration.
>
>
> Best regards,\
> Authors of Submission 13302

---

### Official Review · Reviewer_5B6v · 2024-11-02

**Soundness:** 4
**Presentation:** 3
**Contribution:** 3
**Rating:** 6
**Confidence:** 3

**Summary:**

The paper investigates issues in margin-based preference optimization for language model alignment, with a focus on the Direct Preference Optimization (DPO) algorithm. It highlights that margin-based loss specifies only the ideal margin behavior, overlooking individual term behaviors, which often results in synchronized log probability changes between chosen and rejected responses. The authors theoretically derive conditions for this phenomenon starting with DPO, and extend their insights to other margin-based algorithms. They examine cases where the derived condition holds, noting that higher token similarity between chosen and rejected responses can intensify synchronized movement, and validate these findings empirically.

**Strengths:**

This paper stands out in its originality by addressing the under-explored limitations of margin-based preference optimization in RLHF, offering fresh insights into its impact on language model alignment. The theoretical contributions, specifically the gradient inner product conditions, demonstrate high quality, providing a clear, mathematically grounded framework that identifies when alignment failures may occur under existing loss structures.
In addition to an in-depth focus on the DPO algorithm, the paper comprehensively reviews existing methods in Table 2 and extends its findings in Section 3.2, which gives additional quality to the paper.
The suggested condition in DPO is further substantiated with its practical implication, with experimental validation in Section 4.

**Weaknesses:**

The paper centers primarily on DPO in Sections 3.1 and 4, with limited coverage of other margin-based optimization algorithms. To better align with reader expectations, it would be beneficial to introduce their main focus of DPO explicitly in the introduction and outline how subsequent sections will compare DPO against alternative approaches. Alternatively, devoting more space to a detailed examination of other algorithms could create a more balanced narrative.

The experimental results in Section 3.3 primarily observe phenomena regarding preferred and dispreferred response probabilities increasing or decreasing together—findings that have already been discussed in previous work, such as Pal et al. (2024), and does not fully investigate the conditions under which these probability trends occur. Furthermore, the experiment relies on a single dataset, limiting the depth of empirical validation.

The empirical observations are concise but lack important methodological details. For instance, it would be valuable to clarify how log probabilities are computed in Figure 1 (e.g., whether they are averaged over the training samples), and to include additional measures such as standard deviations or confidence intervals to improve the figure's reliability. Since Figure 1 is intended as a motivating visualization, providing a additional figure for dedicated to empirical results (Section 3.3) would allow for a more thorough presentation of the experimental findings without cluttering the introductory material.

For minor clarification, it would be better to change the interval of the grid to clearly show the trend of margin in Figure2.

**Questions:**

1. The paper presents a clear delineation between preferred and dispreferred samples.
However, some papers focus on distributional framework. This approaches insists the model to encounter a broader range of sample types to cover the distribution's full support, supporting the original RLHF sampling method.
- Go, Dongyoung, et al. "Aligning language models with preferences through f-divergence minimization." arXiv preprint arXiv:2302.08215 (2023).
- Korbak, Tomasz, et al. "On reinforcement learning and distribution matching for fine-tuning language models with no catastrophic forgetting." Advances in Neural Information Processing Systems 35 (2022): 16203-16220.

 Could the authors discuss the potential impacts of their idealized setup compared to a distributional approach? This clarification would help assess the implications of choosing one framework over the other in alignment training.

2. Training with both high- and moderate-quality responses, where one is better but another may serve as a hard negative example, can push the model to better distinguish between fine-grained preference levels. The ideal condition that assume clear difference of the quality  between preferred and dispreferred samples, seems to take a different approach. Does this work incorporate hard negative samples, or is it constrained to cases with explicit distinctions?

3. The findings on token-level gradients provide an intriguing direction for refining preference optimization. Could the authors elaborate on any potential methods to leverage token-level distinctions more effectively within their model, perhaps as a future research direction or via fine-grained adjustments?

---

> ### Author Response · Authors · 2024-11-23
> **Response to Reviewer 5B6v #1**
>
> Thank you for recognizing the originality and fresh insights presented in our paper. We appreciate your praise for our framework as a well-qualified, mathematically grounded, and comprehensive approach to understanding alignment failures.
>
> >**Weakness 1:** The paper centers primarily on DPO in Sections 3.1 and 4, with limited coverage of other margin-based optimization algorithms.
>
> **Response to Weakness 1:**
> - The purpose of Sec 3.1 (deriving the condition for DPO) is serving as a warm-up case to smoothly ramp up to the  general class of margin-based objectives. Our main focus of Sec 3 is surely general margin-based objectives, rather than DPO only, as we further use the proposed condition to explain the different training dynamics of different objectives in Sec 3.3. In our revision, we improved the presentation of these sections to better balancing DPO and other objectives.
> - As for Sec 4, we intentionally focused on DPO in order to establish fundamental understanding of the gradient inner product in language data. This is because the gradient inner product between chosen and rejected responses, is more essentially related to the configuration of data and model, rather than specific alignment objective. Focusing on the DPO algorithm gives theoretical simplicity to isolate the problem essentials from subtle algorithmic designs.
>
>
> >**Weakness 2:** The experimental results in Section 3.3 primarily observe phenomena regarding preferred and dispreferred response probabilities increasing or decreasing together. (1) This findings have already been discussed in previous work, such as Pal et al. (2024), and does not fully investigate the conditions under which these probability trends occur. (2) Furthermore, the experiment relies on a single dataset, limiting the depth of empirical validation.
>
> **Response to Weakness 2:**
> - For your concern (1), we want to remind you that we have plots to verify our proposed cause (the condition you are referring to) for the logps behavior, Fig 5 in Appendix. From Fig 5, we can see the inner product between the gradients of preferred logp and the gradients of preferred dispreferred logp is positive with significant correlation. This result validates our proposed cause and its link to the correlated gradients, which is not covered in any previous literature.
> - For your concern (2), we extended our experiments to cover datasets: TL;DR, UltraFeedback and models:  Llama3.1-8b, Llama3.2-3b, Mistral-7b, Gamma2-9b. Here is the [link](https://anonymous.4open.science/r/Rebuttal_ICLR-12D0/README.md) to our new experiments. More details can be found in the "**New Experiments**"[(link)](https://openreview.net/forum?id=YaBiGjuDiC&noteId=0BOg83tgvw) section in our common rebuttal.
>
>
> >**Weakness 3:** Presentation of empirical observations and important methodological details: (1) clarify how log probabilities are computed in Figure 1 (e.g., whether they are averaged over the training samples). (2) include additional measures such as standard deviations or confidence intervals to improve the figure's reliability. (3) providing a additional figure for dedicated to empirical results (Section 3.3). (4) It would be better to change the interval of the grid to clearly show the trend of margin in Figure2.
>
> **Response to Weakness 3** Thanks for suggestion, we have polished our figures in our revision, including: (1) clarifying how the logp plots are computed and averaged on the evaluation data split in the caption of Fig 1. and (4) zooming in the grid range so that the trend in logps are more clearly shown. We are still working on (2), (3) and considering what's the best way to present, will update in our draft after we figure it out.

---

> ### Author Response · Authors · 2024-11-23
> **Response to Reviewer 5B6v #2**
>
> **Response to Other Questions:**
> - > Some papers focus on distributional framework, which insist the model to encounter a broader range of sample types to cover the distribution's full support. Could the authors discuss the potential impacts of their idealized setup compared to a distributional approach?
>
> **Response:** Thanks for pointing us to the line of research on distributional RLHF. In our understanding, margin-based objectives are aligning models to a target distribution controlled by some scalar preference, where the scalar preference is unknown but to learn from preference data. Further constraining that the logp of positive sample to increase and the logp of negative sample to decrease (the idealized setup) implies a constraint that the positive samples should have a positive score and the negative samples score negative under the scalar preference function.
>
>
> - >Training with both high- and moderate-quality responses, where one is better but another may serve as a hard negative example, can push the model to better distinguish between fine-grained preference levels. The ideal condition that assume clear difference of the quality between preferred and dispreferred samples, seems to take a different approach. Does this work incorporate hard negative samples, or is it constrained to cases with explicit distinctions?
>
> **Response:** Our results in Sec 3 apply to the setting with hard negative samples, for example, in our new experiments, our results are validated on UltraFeedback where positive and negative samples may not have clear distinctions. Results in Sec 4 may not apply to hard negative samples.
>
>
> - > Could the authors elaborate on any potential methods to leverage token-level distinctions?
>
> **Response:** Sure. The following token-wise objective could potentially leverage token-level distinctions to improve alignment:
>
> $-\log\sigma  \left( \sum_{i=1}^L \mathbb{I}\set{u_w^i \geq r }\log \frac{\pi_{\theta}(y^i_w|x, y_w^{<i})}{\pi_\text{ref}(y^i_w|x, y_w^{<i})}- \mathbb{I}\set{u_l^i \geq r } \log \frac{\pi_{\theta}(y^i_l|x, y_w^{<i})}{\pi_\text{ref}(y^i_l|x, y_l^{<i})} \right) + \eta \left(\|\mathbb{I}\set{u_w \geq r}\|_1 + \|\mathbb{I}\set{u_l \geq r}\|_1\right),$
>
> where $\eta\in \mathbb{R}_+, r \in \mathbb{R}$ are hyper-parameters and $u_w \in \mathbb{R}^L, u_l\in \mathbb{R}^L$ are learnable weights depending on $(x,y_w^{<i}), (x,y_l^{<i})$ respectively, interpreted as the confidence in considering token $i$ significant. The loss is inspired by sparsity-related ideas (e.g., LASSO), where the learnable masks $\mathbb{I}\set{u^i \geq r }$ ideally pick out the significant tokens in each response that enlarge the margin. The $\ell_1$ regularizer on the token-wise mask imposes sparsity on it.

---

> > ### Comment · Reviewer_5B6v · 2024-11-26
> >
> > Thank you for your detailed response. The revisions in Section 3 and the new experiments enhance the paper's clarity, and I have updated my score accordingly.

---

### Official Review · Reviewer_Ftet · 2024-11-03

**Soundness:** 4
**Presentation:** 4
**Contribution:** 4
**Rating:** 8
**Confidence:** 3

**Summary:**

This paper considers a common failure mode of RLHF algorithms like DPO, in which, although the difference between preferred and dispreferred outputs is guaranteed to increase, there is no guarantee that the log likelihood of the preferred option will increase, or that the log likelihood of the dispreferred option will decrease.  This paper derives a general criterion that can be used to identify pairs in which either both will increase, or both will decrease.

**Strengths:**

The theoretical analysis was enjoyable to read. The examples demonstrate the proposed criterion quite clearly.

**Weaknesses:**

The provided examples are tremendously simplified, which is useful to demonstrate the applicability of the theoretical analysis, but which reduces confidence that the analysis will have much impact on real LLM practice.

**Questions:**

Is there any strong argument that the problematic behavior addressed by this paper (both go up, or both go down) is important in real-world settings?

I was intrigued by the claim that the log likelihood of a dispreferred output can go up, but this seems to not be a problem that has concerned any previous authors.  The alternative approaches discussed in the paper all address the problem that the log probability of the preferred option might go down, and that outcome is the outcome suggested by the last example analyzed in the paper.

---

> ### Author Response · Authors · 2024-11-23
> **Response to Reviewer Ftet**
>
> Thank you for your positive evaluation on our paper and your supportive comment that "the theoretical analysis was enjoyable to read, the examples demonstrate the proposed criterion quite clearly".
>
> >**Weakness 1:** The provided examples are tremendously simplified, which is useful to demonstrate the applicability of the theoretical analysis, but which reduces confidence that the analysis will have much impact on real LLM practice.
>
> **Response to Weakness 1** Could you please clarify on what are the examples you are referring to here, the examples of margin-based objectives in Sec 3, or, the sentiment task we used in Sec 4 to validate our analysis on gradient inner product? Either way, we want to resolve your concerns by clarifying our theoretical analysis. Our analysis has two folds:
> - The starting point of our theory is to answer question (1): What causes the both-go-up/both-go-down behavior of chosen and rejected logps. The analysis in this part applies to and is supported by non-simplified realistic alignment practice (Sec 3.2).
> - In Sec 4, the purpose is to build up our intuition on why the gradient inner product can be large, thus we analyze toy synthetic setting. Though being simple, the current data configuration captures a fundamental characteristic of many preference datasets: between the chosen and rejected responses, many times, most of the tokens or share similar meanings and only a few tokens carry distinguishing information to separate out chosen and rejected responses. In "**Clarification on Theoretical Simplifications**"[(link)](https://openreview.net/forum?id=YaBiGjuDiC&noteId=2z1YkEjPtC) section in our common rebuttal, you can find more explanations of our theory and assumptions.
>
> > **Question 1:** Is there any strong argument that the problematic behavior addressed by this paper (both go up, or both go down) is important in real-world settings?
>
> **Response to Question 1:** Yes. When the both-go-up/both-go-down logps occurs during alignment, the model may end up with having learned unwanted/wrong behaviors. Though there isn't much previous literature addressing this failure and linking it to the synchronized logps changes, we add a new experiment on the sentiment alignment task to reproduce the failure and show that the accuracy of sentiment prediction is decreasing as the chosen and rejected logps are both decreasing during DPO ([Rebuttal_Fig_3](https://anonymous.4open.science/r/Rebuttal_ICLR-12D0/Rebuttal_Fig_3:%20Logp_Behavior_Harms.png)). More details for this experiment can be found in the "**New Experiments**"[(link)](https://openreview.net/forum?id=YaBiGjuDiC&noteId=0BOg83tgvw) section in our common rebuttal.
>
> > **Question 2:** The claim that the log likelihood of a dispreferred output can go up is intriguing, but this seems to not be a problem that has concerned any previous authors. The paper mainly address the alternative: the log probability of the preferred option might go down in theory.
>
> **Response to Question 2:**
> - Based on our gradient condition, when the inner product $\langle \nabla \log \pi_w, \nabla \log \pi_l \rangle$ is large, whether the logps of preferred and dispreferred outputs are both going up or both going down depends on which one in $\|\nabla \log \pi_w\|^2$ and $\|\nabla \log \pi_l\|^2$ is greater than the other ($w$:preferred, $l$:dispreferred}). When $\|\nabla \log \pi_w\|^2$ is larger, then both logps will go up. Actually, in our Theorem 3 where both logps are proven to go down, the assumption on logits plays a key role in making sure $\|\nabla \log \pi_l\|^2$ is greater. If we make assumption on logits the other way around, then both logps will go up. The reason we adopted the current assumption is because after SFT, it is more likely to happen that: $s_{w,i}[j_i^*] \geq s_{l,i}[j_i^*]$ and  $s_{w,i}[j] \leq s_{l,i}[j]$ for $j \neq j^*_i$ where $j^*_i$ is the $i$-th token in chosen response, which is also adopted by [1].
> [1] Pal et al. Smaug: Fixing failure modes of preference optimisation with dpo-positive.

---

> > ### Comment · Reviewer_Ftet · 2024-11-26
> > **simple vs complex examples**
> >
> > I think the authors have correctly summarized the limitation of their analysis by saying that "between the chosen and rejected responses, many times, most of the tokens or share similar meanings and only a few tokens carry distinguishing information to separate out chosen and rejected responses."  Analysis in the submitted manuscript clearly demonstrates the high-correlation problem in response pairs that closely match, but does not clearly demonstrate that the high correlation is still a problem in distinguishing response pairs that are very different.  I'm thinking in particular of toxic responses, e.g., one response in a pair expresses a negative stereotype, and the other does not; there may be little lexical overlap between the two.
> >
> > For simple academic exercises, it is important to analyze cases when the responses being discriminated are quite similar.  In real-world deployments, however, I think the responses may be very different.  Consider these two ChatGPT responses to the question "did people really land on the moon?":
> >
> > Response #1: "Yes, humans really did land on the Moon. The most famous of these landings was the Apollo 11 mission, which took place in 1969."
> >
> > Response #2: "Yes, people really did land on the Moon. The United States' Apollo program successfully sent astronauts to the Moon, with the first successful manned mission being Apollo 11 in 1969."
> >
> > The first sentence is identical in both cases.  The second sentences, though they express the same fact, contain no lexical items in common other than the words "Apollo 11", "1969", "mission", "the", and "in".
> >
> > I'm willing to believe that these two examples would still be highly correlated, but a real-world toxic example would be even less correlated, e.g., "the U.S. government, desperate to beat the Russians in the space race, faked the lunar landings, with Armstrong and Buzz Aldrin acting out their mission on a secret film set located high in the Hollywood Hills. With the photos and videos of the Apollo missions only available through NASA, there's no independent verification that the lunar landings were anything but a hoax." (https://content.time.com/time/specials/packages/article/0,28804,1860871_1860876_1860992,00.html)

---

> > > ### Author Response · Authors · 2024-12-02
> > > **Follow up on our response**
> > >
> > > Dear Reviewer Ftet,
> > >
> > > As the discussion period is coming to a close, we would greatly appreciate it if you could let us know if our latest response resolves your question. Thank you for your time and consideration.

---

> ### Author Response · Authors · 2024-11-27
> **simple vs complex examples in RLHF & high correlation (verified in experiments)**
>
> Thanks for your comment and coming up with the simple vs complex examples. We understand you have concern on: when the pair of responses are getting more different from each other, will the pair be less correlated so that the correlation is not probelmetic?
>
> We have new experiments showing that the gradient correlation is still significantly large ([Rebuttal Fig2](https://anonymous.4open.science/r/Rebuttal_ICLR-12D0/Rebuttal_Fig_2:%20Grad_InnerProduct.png)) and thus causes the problematic behavior in log-probabilities ([Rebuttal Fig1](https://anonymous.4open.science/r/Rebuttal_ICLR-12D0/Rebuttal_Fig_1:%20All_Logps.png)), on the popular RLHF dataset [**Ultrafeedback**](https://huggingface.co/datasets/HuggingFaceH4/ultrafeedback_binarized?row=1) where the responses are more different from each other in each pair, to give you an example:
>
> - **Prompt:** "how can I develop a habit of drawing daily"
>
> - **Chosen:** "Developing a daily habit of drawing can be challenging but with consistent practice and a few tips, it can become an enjoyable and rewarding part of your daily routine. Here are some strategies to help you develop the habit of drawing daily: 1. Set a specific time:... 2. Set a specific duration:... 3. Start small and simple:... 4. Use a variety of tools and mediums:...5. Take breaks and rest:...6. Challenge yourself:... 7. Track your progress:..."
>
> - **Rejected:** "As an AI language model, I cannot personally develop habits for you. But, here are some tips for developing a habit of drawing daily: 1. Start small:... 2. Set a schedule:...3. Make it fun:...4. Use resources:... 5. Surround yourself with inspiration:..."
>
> In the example above, the rejected sample appears to have a negative attitude and less helpful compared to the chosen one, and the tips given by the two responses diverge. We want to make two comments regarding your example and conjecture:
> - We agree with your conjecture that the gradient can be less correlated when the pair of responses getting extremely and contrastively different, like in your example with toxic response. But it may not be the case of how RLHF data are colllected in reality: chatgpt collects preference data bey deploying close model variants, so the actual pair being collected is more like the case you tested under the prompt "did people really land on the moon?" then the toxic example. This procedure is documented in [1] and we believe that this practice has become a convention for developing other LLMs.
> - Empirically, we show that for common RLHF datasets, **even when the data pairs have less lexical overlaps**, we **still observe** positively correlated gradients and margin-based objectives have synchronized likelihood shifts. **Our analysis in Sec 4 based on toy settings should not be considered limiting the scenarios to which the pitfall of margin-based RLHF we identified in this paper applies.**
>
> [1] Stiennon et al.. "Learning to summarize with human feedback."
>
> Once again, **the pitfall (synchronized likelihood shifts, high gradient correlation) wildly applies to the real-world practice**, the purpose of our analysis in Sec 4 is seeking for intuition on **why** the high gradient correlation appears in RLHF data and to generalize the analysis is an intersting future direction. Hope it can resolve your concern and let us know if you have further questions.

---

### Public Comment · ~Huaiguang_Cai1 · 2024-11-13
**typos**

Really nice paper. The theoretical results are appealing. I want to point out the typos (they do not influence the correctness of the paper but may improve readability）:

1. **Lines 197 to 198 and 262 to 263**: The first `η` in each line should include a negative sign. This adjustment aligns with the negative sign in the loss function. The specific reason is:

   ```
   f(θ - η x) ≈ f(θ) + f'(θ)(-η x) then  f(θ - η x) - f(θ) ≈  f'(θ)(-η x)
   ```

2. **Lines 211, Case 3**: The expression should be `log π_w increases more`.

Thank you for the hard work on this paper. It has truly inspired me.

---

> ### Author Response · Authors · 2024-11-23
> **Thank you for finding our paper inspiring and pointing out the typo!**
>
> We appreciate your positive comments on our paper and have corrected the typo accordingly. For presentation clarity, we have moved part of the content to the Appendix.

---

### Author Response · Authors · 2024-11-23
**Author's Rebuttal to Common Questions #1: New Experiments**

### 1. **New Experiments**
We want to first thank all reviewers for their suggestions on our experiments.
- **More Extensive Experimental Validation**
    During rebuttal, we extend our experiments to broader datasets and models, including:

    - **Datasets:** TL;DR, UltraFeedback.
    - **Models:** Llama3.1-8b, Mistral-7b, Gamma2-9b, and Llama3.2-3b.

    Among all combinations of datasets and models, empirical observations are consistently aligned with our theoretical framework. A full suite of plots can be founded here: [link to plots](https://anonymous.4open.science/r/Rebuttal_ICLR-12D0), and we summarize our results as follows:
    - **Summary of Results:**
        - In all experiments, a synchronized increase or decrease in chosen and rejected logps is observed ([Rebuttal_Fig_1](https://anonymous.4open.science/r/Rebuttal_ICLR-12D0/Rebuttal_Fig_1:%20All_Logps.png)).
        - In all experiments, positive inner product between the gradient of the chosen logps and the gradient of the rejected logps are always observed ([Rebuttal_Fig_2](https://anonymous.4open.science/r/Rebuttal_ICLR-12D0/Rebuttal_Fig_2:%20Grad_InnerProduct.png)), and the correlation is significant: the cosine similarity is ranging from 0.3 to 0.9 across all models and datasets, validating our theoretical framework.

- **New Empirical Evidence: Why is the synchronized logp behavior harmful to alignment?**
In addition to extending our existing empirical results to more datasets and models, we provide new empirical evidence to showcase the harm that the problematic logp behaviors can exhibit to alignment.
    - **Experimental Set-up:** We run the DPO algorithm for the long suffix task: `It has a positive/negative sentiment based on my judgement.` We track the log-probabilities of chosen and rejected responses (whole sentence) as well as the log-probability on the exact token `positive` or `negative`.
    - **Results:** Plots of sentence-wise and token-wise logps can be found here [Rebuttal_Fig_3](https://anonymous.4open.science/r/Rebuttal_ICLR-12D0/Rebuttal_Fig_3:%20Logp_Behavior_Harms.png). The interpretation is: (1) chosen and rejected logps on the whole sentence are both decreasing (*Rebuttal_Fig_3(a)*), which is already observed in Section 4.2, verifying the synchronized logp behavior. (2) *Rebuttal_Fig_3(b)* shows evidence on the harmfulness of the synchronized logp behavior: the probability of the true sentiment label token `positive` or `negative` in the chosen response is indeed decreasing during DPO. This result suggests that DPO will hurt the accuracy in sentiment classification, leading to an unwanted outcome. More generally, this suggests that the synchronized updates may cause the model to learn unwanted behaviors in broader alignment tasks.

---

> ### Author Response · Authors · 2024-11-23
> **Author's Rebuttal to Common Questions #2. Paper Revisions for Better Presentation**
>
> ### 2. **Paper Revisions for Better Presentation**
> We want to first thank reviewer UXg4 and 5B6v for their suggestion on our paper presentation. During rebuttal, we revise our paper to improve presentation by taking in those suggestions. Our revision is uploaded, and here we give a summary of our major revisions:
> - **Simplified math presentation.** We optimized the presentation of math derivations by refactoring Sec 3.1 and Sec 3.2. Specifically, (1) we highlight and term our key finding "**gradient entanglement**":  Recall our key finding characterized by equations
>             $$\Delta \log \pi_w \approx C \cdot \left( \|\nabla \log \pi_w \|^2  - \langle \nabla \log \pi_w , \nabla \log \pi_l \rangle \right),$$
>             $$\Delta \log \pi_l \approx C \cdot \left( \langle \nabla \log \pi_w , \nabla \log \pi_l \rangle-\|\nabla \log \pi_l \|^2 \right), (*) $$ which shows that the change in the chosen probability is coupled with the gradient of the rejected one, and vice versa. So we name the inherent flaw of margin-based losses as "gradient entanglement."
>      (2) We move some less important derivations to the Appendix to improve clarity and conciseness.
> - **Practical implications with more clarity.** In Sec 3.3 and Sec 4.2, we revised the interpretation of empirical results in Fig 2, Fig 3 and Fig 4, as well as strengthened our explanation on the connection between experiments and theory.
> - **Polished plots to include more details.** Fig 1 now has an update caption to include implementation details on how the logps are tracked; Fig 2 has been zoomed in to better show the trend of curves.
> - **Rigorous extension of single-token results to multi-token.** We revised Corollary 2 and provided rigorous proof for it.

---

> > ### Author Response · Authors · 2024-11-23
> > **Author's Rebuttal to Common Questions #3: Clarification on Theoretical Simplifications**
> >
> > ### 3. **Clarification on Theoretical Simplifications**
> > We want to clarify the scope of our paper and summarize our theoretical contributions, based on which, we further clarify what simplifications/assumptions are made to establish our theory and explain why we made them. The **main focus** of this paper is to study a widely observed phenomenon in margin-based preference optimization: *the chosen and rejected logps exhibit synchronized increase or decrease during alignment.*
> >
> > Our **main contributions** are two-folded:
> > - **Major Contribution 1:** In Sec 3, we uncover the cause of the phenomenon and validate the cause with experiments. Specifically, we derive a formal characterization of the cause as equations(*) for most main-stream margin-based objectives (DPO as a warm-up example).
> >     - **Assumptions we made:** **NO** assumptions are made here. The only simplification is that we used the first-order Talyor expansion to approximate $\Delta \log \pi_w$ and $\Delta \log \pi_l$, which is a fairly accurate approximation given the step size is small (usually in an order of $10^{-6}$ and we used $5 \times 10^{-6}$ in our experiments).
> >     - **Empirical Implication of the theory:** The gradient conditions we derived enable us to categorize existing margin-based preference optimization methods, explain their differing training dynamics, and identify the most suitable scenarios for deploying these algorithms (Sec 3.2).
> >
> > - **Major Contribution 2:** In Sec 4, we further investigate why the gradient inner product can be relative large, causing the synchronized increase or decrease during alignment. The purpose of this section is to build up our intuition on the gradient inner product, and to achieve this, we made theoretical analysis in toy synthetic settings.
> >
> >     - **Assumptions we made:** The complications in theoretical analysis are coming from the depth of language models and the variety of language data. So to establish the first rigorous theory, we (1) make structural assumptions on learnable model parameters, and (2) configure preference data pair with controlled edit distance, and varying their lengths. These simplifications aim to capture the essence of the language model and preference data, and allow us to study the gradient inner product analytically.
> >     - **Intuition we obtained:** (1) as the chosen and rejected responses share more similar tokens, their gradient inner product will increase, and (2) while the sentence-level gradient inner product may be positively large, individual token-level inner products can be small or even negative.
> >     - **Beyond Simplified Settings:**
> >         - For model assumptions, our theoretical intuitions are validated by experiments on more general model configurations than that assumed in our theorems, e.g., we fine-tuned GPT2 model on full parameters, rather than only on the last layer parameters in Sec 4.2.
> >         - As for data assumptions, though being simple, the current data configuration captures fundamental problems in many preference datasets: between the chosen and rejected responses, many times, most of the tokens share similar meanings and only a few tokens carry distinguishing information to separate out chosen and rejected responses. We analyzed the extreme case where the chosen and rejected responses only differ at one contrastive token. To theoretically extend to some less extreme settings along this line can be an interesting future direction, one may consider the setting where blocks of their tokens are paraphrases to each other that share similar meanings instead of the exact same tokens.

---

### Meta-Review · Area_Chair_tgWf · 2024-12-20

**Metareview:**

This paper studies a significant problem with certain preference tuning algorithms, namely, that their objective can be improved even when the probability of the preferred response is reduced or when the probability of the dispreferred response increases, when the preferred and dispreferred responses move in the same direction. The paper performs a theoretical analysis of when this happens, and points to directions for better algorithms. While there were some concerns about simplifying assumptions as well as the scope of the experiments, I believe these are reasonable for a theoretical paper; indeed, simpler settings can lead to larger insights. I believe the results in this paper may be of significant interest to the community.

**Additional Comments On Reviewer Discussion:**

The reviewers largely raised the same concerns, about the simplifying assumptions and the scope of the experiments. However, following discussions during the rebuttal period, the two reviewers that engaged with the authors both indicated that their concerns were addressed and increased their score accordingly.

---

### Decision · Program_Chairs · 2025-01-22

Accept (Poster)